# An optimized Nurr1 agonist provides disease-modifying effects in Parkinson's disease models

Woori Kim[1,2,5], Mohit Tripathi[3,5], Chunhyung Kim[1,2], Satyapavan Vardhineni[3], Young Cha [1,2], Shamseer Kulangara Kandi[3], Melissa Feitosa[1,2], Rohit Kholiya[3], Eric Sah [1,2], Anuj Thakur[3], Yehan Kim[1,2], Sanghyeok Ko[1,2], Kaiya Bhatia[1,2], Sunny Manohar[3], Young-Bin Kong[1,2], Gagandeep Sindhu[3], Yoon-Seong Kim [4], Bruce Cohen [1], Diwan S. Rawat [3] ✉ & Kwang-Soo Kim [1,2] ✉

The nuclear receptor, Nurr1, is critical for both the development and maintenance of midbrain dopamine neurons, representing a promising molecular target for Parkinson's disease (PD). We previously identified three Nurr1 agonists (amodiaquine, chloroquine and glafenine) that share an identical chemical scaffold, 4-amino-7-chloroquinoline (4A7C), suggesting a structure-activity relationship. Herein we report a systematic medicinal chemistry search in which over 570 4A7C-derivatives were generated and characterized. Multiple compounds enhance Nurr1's transcriptional activity, leading to identification of an optimized, brain-penetrant agonist, 4A7C-301, that exhibits robust neuroprotective effects in vitro. In addition, 4A7C-301 protects midbrain dopamine neurons in the MPTP-induced male mouse model of PD and improves both motor and non-motor olfactory deficits without dyskinesia-like behaviors. Furthermore, 4A7C-301 significantly ameliorates neuropathological abnormalities and improves motor and olfactory dysfunctions in AAV2-mediated α-synuclein-overexpressing male mouse models. These disease-modifying properties of 4A7C-301 may warrant clinical evaluation of this or analogous compounds for the treatment of patients with PD.

Parkinson's disease (PD), affecting 2–3% of the population over the age of 65, is the second most common neurodegenerative disorder, after Alzheimer's disease[1–3]. Selective degeneration of midbrain dopamine neurons (mDANs) in the substantia nigra and neurotoxic α-synuclein (αSyn) accumulation and aggregation in Lewy bodies are hallmark pathological features of PD. Manifestation of PD is thought to be caused by the combined action of genetic and environmental factors via multiple interacting pathways including mitochondrial dysfunction, oxidative stress, neuroinflammation, and dysregulated protein degradation/autophagy[3–7]. Presently, dopamine (DA)-replacement therapy (e.g., L-DOPA) is the gold standard treatment. While it significantly improves PD patients' quality of life, the therapeutic window without unacceptable side effects, such as dyskinesia, and the degree of benefit both decrease over time[8,9], and there is no treatment that can halt or slow disease progression. Therefore, there is a great unmet need to develop novel disease-modifying treatment for PD[10,11].

During the last several decades, mDANs' transcriptional regulatory cascades have been comprehensively studied[12]. Two major

[1]Department of Psychiatry, McLean Hospital, Harvard Medical School, Belmont, MA 02478, USA. [2]Molecular Neurobiology Laboratory, Program in Neuroscience, McLean Hospital, Harvard Medical School, Belmont, MA 02478, USA. [3]Department of Chemistry, University of Delhi, Delhi 110007, India. [4]Institute for Neurological Therapeutics, Rutgers University, Piscataway, NJ 08854, USA. [5]These authors contributed equally: Woori Kim, Mohit Tripathi.
✉e-mail: dsrawat@chemistry.du.ac.in; kskim@mclean.harvard.edu

mDAN developmental pathways (i.e., Sonic hedgehog-FoxA2 and Wnt1-Lmx1A) converge to induce Nurr1[13–15], highlighting Nurr1 as a master regulator for mDANs. Indeed, Nurr1 knockout (KO) and conditional KO mouse studies demonstrated that Nurr1 is essential not only for the development[16] but also for the maintenance of adult mDANs[17]. In addition, Nurr1 was shown to protect mDANs from neuroinflammation-induced death by suppressing the expression of neurotoxic pro-inflammatory genes[4]. Furthermore, Nurr1 expression has been reported to be significantly diminished in the brains of both aged and sporadic PD patients[18–20].

Nurr1 belongs to the nuclear receptor NR4A subfamily, suggesting that its transcription function can be modulated by synthetic and/or endogenous ligands[21]. In addition, Nurr1 can function as a transcriptional activator and as a repressor, depending on the cellular context (e.g., in mDANs[16] and microglia[4], respectively). To begin to address whether Nurr1 might be a molecular target for disease-modifying treatment of PD[22], we established high throughput assay systems and previously screened a library of US-FDA approved drugs. We identified three drugs, i.e., amodiaquine (AQ), chloroquine (CQ), and glafenine, which significantly enhance the transcriptional activity of Nurr1 through direct binding to its ligand binding domain (LBD)[23]. All three drugs shared an identical chemical scaffold *viz.* 4-amino-7-chloroquinoline (4A7C), prompting us to hypothesize that this 4A7C motif represents a structure-activity relationship (SAR) that could be used to identify better 4A7C-derivatives with higher potency and lesser side effects through SAR-based medicinal chemistry[24]. In line with this hypothesis, recent works from Munoz-Tello et al. [25] and Willems et al. [26,27] showed AQ and CQ and their derivatives can work as Nurr1 agonists. Willems et al. [26] also reported that removal of the 7-chloro or the 4-amino group of the 4A7C pharmacophore leads to the loss of activity, further supporting our hypothesis that 4A7C is a valid pharmacophore for designing potent Nurr1 agonists. Toward this end, we performed a systematic medicinal chemistry effort using CQ as the starting compound because it is relatively safe and is still widely used to treat human diseases such as malaria and autoimmune diseases[28]. We characterized multiple (>570) 4A7C-derivatives using biochemical, molecular, and cellular assays and identified a final candidate, 4A7C-301, that exhibits robustly higher potency and provides mechanism-based neuroprotective effects in both in vitro and in vivo models of PD, separately using both an environmental (the mitochondrial complex I inhibitor 1-methyl-4-phyenylpyridinium (MPP$^+$)) and a genetic (αSyn) risk factor models of PD. Unlike L-DOPA or CQ, 4A7C-301 significantly improved both motor and non-motor dysfunctions without side effects such as any dyskinesia-like behaviors or autophagy inhibition. Our data demonstrate proof of principle that optimized Nurr1 agonist(s) may provide disease-modifying treatment for both sporadic and familial PD.

## SAR-based optimization of Nurr1 agonists

Previously, our group designed and synthesized potent anti-malarial hybrids active against drug-resistant *P. falciparum* strains[29–32]. These hybrids were based on linking the 4A7C pharmacophore to other antimalarial pharmacophores (such as triazines/pyrimidines) through a flexible alkyl chain linker, similar to that of CQ. We speculated that some of these 4A7C-derivatives would activate Nurr1's transcription function, if 4A7C is a valid SAR. Indeed, using our established Nurr1 reporter assays[23,33], we found that ~20% of these derivatives exhibited detectable activation of Nurr1 function. We also observed that when the 4A7C pharmacophore is linked to a substituted triazine or a pyrimidine ring, it showed prominent Nurr1-activation (EC$_{50}$/max fold induction up to 1.10 μM/4.76 fold for 4A7C-triazines and up to 121.22 nM/5.08 fold for 4A7C-pyrimidines; see Fig. 1a). The 4A7C-pyrimidine hybrid 4A7C-101 emerged as the first lead compound activating Nurr1 with lower EC$_{50}$ value (121.22 nM) than CQ (50.25 μM) and was selected for further optimization (Fig. 1a; Supplementary

Table 1). An exhaustive lead optimization programme was then carried out to assess the effects of structural modifications at the three major modification sites of 4A7C-101 (Fig. 1b), resulting in the generation of additional 4A7C-derivatives (~470). This SAR analysis revealed important points for further optimization of the 4A7C-pyrimidine derivatives (see Fig. 1b). First, for linker, a flexible and short (C2-C3) diamino aliphatic chain linker connecting the 4A7C to the pyrimidine ring showed good activity. When piperazine was used as linker (rigid, no free -NH at quinoline's 4$^{th}$ position), activity was lost, indicating the essentiality of the 4-amino group of the quinoline ring. The other −NH of linker (linking the pyrimidine ring) can be substituted or even removed, with preservation of activity. Compounds with −CH$_2$-1,3-benzodioxole or −CH$_2$-thiophene substituent displayed good activity (see compounds 4A7C-501 to 4A7C-508 in Supplementary Table 1). A hybrid aliphatic chain-piperazine linker also displayed good potency (see compounds 4A7C-201 to 4A7C-208 in Supplementary Table 1). Replacement of amodiaquine's −OH with −F to form fluoro-amodiaquine-pyrimidines led to loss of activity, indicating the importance of the AQ's phenolic hydroxyl group for Nurr1 activity (which is also in agreement with Merk et al. [26]). Second, for pyrimidine ring, pyrimidine regio-chemistry also plays a role in activity, but the activity depends on the substituents on the pyrimidine ring and linker. When quinazoline ring (a fused pyrimidine) was used instead of pyrimidine, activity was preserved, but a slight dip was observed. Third, for pyrimidine ring substituents, substituents with free −NH group (such as amino-alcohols) decreased activity and carbocyclic amine substituents were important for good activity. In few cases, molecules with -Cl substituents (intermediates to final compounds) also showed activity.

It was observed that substituents in the pyrimidine ring have a major effect on the overall Nurr1 activity of the 4A7C-pyrimidine derivatives and hence we further put our efforts on optimizing the substituents in the pyrimidine ring. Compound 4A7C-102, analogous to 4A7C-101 but having an N-ethyl piperazine substituent in the position-2 of the pyrimidine ring was the second-best compound identified thus far (3.68 fold activation at EC$_{50}$ of 143.13 nM). We then considered the possibility of replacing the −Me group at the 4$^{th}$ position of the pyrimidine ring of 4A7C-101 with other substituents, initially keeping the piperidine substituent of 4A7C-101 constant, to understand its role in Nurr1 activity. 4A7C-101 analogues in which the −Me substituent was replaced with a −Ph substituent, a −CF$_3$ substituent, a −Cl substituent or an −H atom displayed diminished Nurr1 activation. We also studied the influence of a shorter C2 linker and pyrimidine regiochemistry and found that the compound 4A7C-103 with a different pyrimidine regiochemistry and having a −Cl substituent and an N-ethyl piperazine substituent retained diminished Nurr1 activity. Furthermore, the analogous compound 4A7C-104 having a single point of difference vis-à-vis C2 linker instead of a C3 linker displayed better Nurr1 EC$_{50}$ than 4A7C-103 (see supplementary Table 1). Next, we studied the influence of di-substitution with different cyclic amines at 4$^{th}$ and 6$^{th}$ position of the pyrimidine ring and keeping the linker chain length of 2 carbons (i.e. 1,2-diaminoethane linker). As a first step, to understand the effect of ring size and bulkiness of cyclic amines on Nurr1 activation, the pyrimidine was di-substituted with pyrrolidine (5-membered ring), piperidine (6-membered ring) and azepane (7-membered ring). The 6-membered piperidin-1-yl substituted compound 4A7C-302 displayed the maximum activation (5.19 fold at EC$_{50}$ of 950.63 nM). In the next step, other 6-membered cyclo-amine substituents *viz.* morpholine, *N*-methyl piperazine and *N*-ethyl piperazine were introduced as di-substituents on the pyrimidine ring. Among them, 4A7C-301, with point of attachment to the linker being the 2nd-position of the pyrimidine ring and having two 4-ethyl-piperazin-1-yl substituents at the 4$^{th}$- and 6$^{th}$-position of the pyrimidine ring, displayed the highest Nurr1 activation along with a significantly lower EC$_{50}$ value compared to CQ. Its regiomer compound 4A7C-303 also displayed good activity, but with lower fold induction than 4A7C-301.

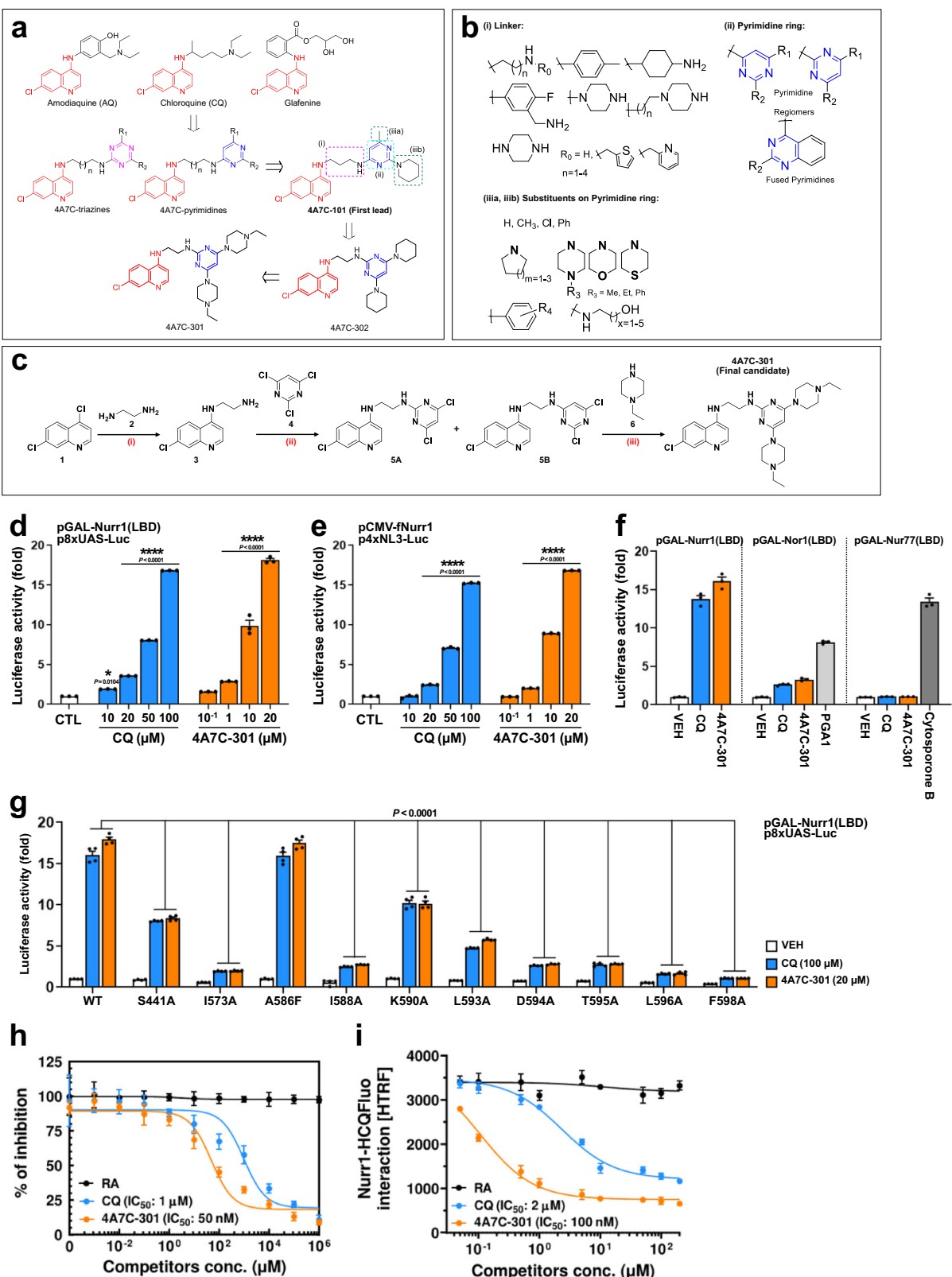

We also kept the activity imparting 4-ethyl-piperazin-1-yl substituent constant at the 4[th] position and varied the cyclic amine substituents at the 6[th] position with piperidine, morpholine and *N*-methyl-piperazine (derivatives 4A7C-304, 4A7C-305 and 4A7C-306, respectively). Although these derivatives displayed good Nurr1 activities (in the order: *N*-methyl piperazine > morpholine > piperidine, the derivative 4A7C-301 with highest fold-induction (18.12 fold) and significantly

lower EC$_{50}$ value (6.53 μM) than CQ (50.25 μM) was selected as a preclinical candidate for further studies (Fig. 1c; Supplementary Table 1). Brain permeability of 4A7C-301 was assessed by analysis of blood plasma and brain homogenates at different time points after oral administration in SD rats (20 mg/kg). 4A7C-301 penetrated into the brain well and was consistently maintained in the brain (Supplementary Fig. 1a–c).

**Fig. 1 | Identification of 4A7C-301 as an optimized agonist for Nurr1 activation.** **a, b** Development of 4A7C-Pyrimidine conjugates as potent Nurr1 agonists from CQ based on the structure-activity relationship (SAR) sharing an identical scaffold, 4-amino-7-chloroquinoline (4A7C; indicated in red color); (i), (ii), (iiia)/(iiib) denote the sites of structural modifications on 4A7C-pyrimidines. **c** Synthetic scheme for 4A7C-301; Reaction conditions: (i) neat, 120 °C, 8 h, 90%; (ii) triethylamine, THF, 60 °C, 12 h, 28 % (5A), 53% (5B); (iii) neat, sealed-tube, 110 °C, 5 h, 65%. **d, e** Luciferase assays using Nurr1-LBD (**c**) or full-length Nurr1 (**d**) in SK-N-BE(2)C cells. *$P < 0.05$, ****$P < 0.0001$ compared to control (CTL), two-way ANOVA, Dunnett's post-hoc test; $n = 3$ biologically independent samples per group. Data are mean ± s.e.m. **f** Luciferase assay determining the effect of CQ (100 μM) and 4A7C-301 (20 μM) using LBD of NR4A subfamily members in SK-N-BE(2)C cells. Prostaglandin A1

(PGA1, 10 μM) and cytosporone B (10 μM) were used as positive controls of Nor1- and Nur77-activation, respectively. $n = 3$ biologically independent samples per group. Data are mean ± s.e.m. **g** Luciferase assay using point-mutants on potential Nurr1-LBD binding residues in SK-N-BE(2)C cells. $P < 0.0001$ compared to CQ or 4A7C-301 treated wild-type (WT), two-way ANOVA, Dunnett's post-hoc test; $n = 4$ biologically independent samples per group. Data are mean ± s.e.m. **h** Competition analysis of CQ or 4A7C-301 with [³H]-CQ for binding to Nurr1-LBD. Retinoic acid (RA) was used as a negative control. $n = 3$ biologically independent samples per group. Data are mean ± s.d. **i**, Time-resolved fluorescence resonance energy transfer (TR-FRET) assay showing dose-dependent competition of CQ or 4A7C-301 with fluorescence-labeled hydroxy-CQ (HCQFluo) for binding to Nurr1-LBD. $n = 3$ biologically independent samples per group. Data are mean ± s.d.

## Neuroprotection by 4A7C-301 via Nurr1 activation in vitro

To assess the potency of 4A7C-derivatives in promoting Nurr1's transcriptional activity, we initially used the human neuroblastoma SK-N-BE(2)C cell line because it is ideal for high throughput screening due to its high transfection efficiency[23,33]. 4A7C-301 increased transcriptional activities of both Nurr1-LBD (Fig. 1d) and full-length Nurr1 (Fig. 1e) in a dose-dependent manner, with a higher potency than CQ (EC50 was 7-8 and 50-70 μM, respectively). We next analyzed the selectivity of CQ and 4A7C-301 toward other NR4A subfamily members (Nur77 and Nor1). As shown in Fig. 1f, in SK-N-BE(2)C cells, CQ and 4A7C-301 induced Nurr1's transcriptional activity with the greatest potency (13-14 fold) among NR4A members. We also found that both compounds significantly activated Nor1's transcriptional activity but with lower efficiency (3-4 fold). In contrast, they did not activate Nur77's transcriptional activity while the Nur77 agonist, cytosporone B[34], robustly increased it. Together, these data suggest that CQ/4A7C-301 selectively activates the transcriptional activity of Nurr1, but not Nur77, over Nor1. Although SK-N-BE(2)C cells express the tyrosine hydroxylase (TH) gene, they are not of mDAN origin. Thus, we compared the transcriptional potencies of 4A7C-301 and CQ in the dopaminergic cell line N27-A, which originated from *rat* mDANs[35,36]. Remarkably, 4A7C-301 exhibited approximately 50 times higher potency than CQ in N27-A cells (EC50 was ~0.2 and ~10 μM, respectively; Supplementary Fig. 1d,e). To address whether 4A7C-301 functions via interacting with Nurr1-LBD, we examined the effects of specific Nurr1-LBD mutations at the perturbed residues based on our previous NMR titration studies[23,33,37]. Although these studies demonstrated that AQ[23], CQ[37], and PGA1/PGE1[33] interact with Nurr1-LBD, specific interacting residues were significantly different among these agonists. Having selected 4A7C-301 among CQ derivatives, we chose residues S441, I573, A586, I588, K590, L593, D594, T595, L596 and F598 that were previously identified as CQ-interacting sites[37]. Significant reduction of transcriptional activity was observed in the constructs with mutations at most (I573, I588, L593, D594, T595, L596, and F598) but not in all these residues in similar patterns with CQ and 4A7C-301 (Fig. 1g). Those findings confirm these sites as critical for activation and interaction between Nurr1 and CQ/4A7C-301, and further reveal that both CQ and 4A7C-301 activate Nurr1 via direct binding to Nurr1-LBD. Mutations at certain residues (e.g., S441, K590, L593, and D594) did not affect the basal transcriptional activity but decreased CQ- or 4A7C-301-induced Nurr1 transcriptional activity compared to the wild-type reporter construct. When these mutant constructs were further tested by treating the transfected cells with 4A7C-301 as well as with PGA1 and AQ, Nurr1 transcriptional activation by 4A7C-301, but not those by AQ or PGA1, was significantly diminished, supporting the conclusion that these residues are 4A7C-301 (and CQ)-specific (Supplementary Fig. 2). Together, these site-directed mutagenesis analyses suggest that different synthetic and native agonists of Nurr1 interact with shared as well as distinct residues within the Nurr1-LBD and specifically activate Nurr1's transcriptional function. In addition, radioligand binding/competition assay using [³H]-CQ demonstrated that CQ and 4A7C-301 can

compete with [³H]-CQ for binding to Nurr1-LBD with IC50s of 1.03 ± 0.61 μM and 48.22 ± 22.05 nM, respectively (Fig. 1h). Furthermore, a ligand binding assay using a time-resolved fluorescence resonance energy transfer (TR-FRET) system along with fluorescent-labeled hydroxy-CQ (HCQFluo), CQ and 4A7C-301 competed with HCQFluo with IC50s of 2.33 ± 0.52 μM and 107.71 ± 14.14 nM, respectively (Fig. 1i). These results indicate that 4A7C-301 has 20-fold higher binding affinity for Nurr1-LBD than CQ, consistent with its higher potency. When tested in another murine dopaminergic cell line, MN9D[38], 4A7C-301 again exhibited much higher potency (~50-fold) than CQ to activate Nurr1 (Supplementary Fig. 3a,b).

To determine whether CQ and 4A7C-301 can induce neuroprotective effects, we treated N27-A and MN9D cells that were exposed to the mitochondrial complex I inhibitor, 1-methyl-4-phyenylpyridinium (MPP⁺) with CQ or 4A7C-301. CQ and 4A7C-301 each significantly decreased MPP⁺-induced cytotoxicity in a dose-dependent manner, and 4A7C-301 showed its maximal efficiency at ~20 times lower concentration than CQ in both N27-A (Supplementary Fig. 1f, g) and MN9D cells (Supplementary Fig. 3c, d). To confirm that these neuroprotective effects of CQ/4A7C-301 are Nurr1-dependent, we tested cell viability and cytotoxicity against MPP⁺ in Nurr1 overexpressing (OE) or knocked down (KD) conditions in N27-A and MN9D cells. Nurr1 OE potentiated the neuroprotective effects of CQ (100 μM) and 4A7C-301 (1 μM) against MPP⁺ compared to control (Fig. 2a, b; Supplementary Fig. 3e, f). In contrast, Nurr1 KD not only significantly reduced cell viability in the absence of MPP⁺ (approximately 20%) but also completely abrogated the neuroprotective effects of CQ and 4A7C-301 against MPP⁺ in both N27-A (Fig. 2a, b) and in MN9D cells (Supplementary Fig. 3e, f), strongly suggesting that neuroprotection by CQ and 4A7C-301 occurs via Nurr1 activation. We next investigated whether mitochondrial dysfunction and/or oxidative stress are involved in cell viability. Treatment of N27-A cells with MPP⁺ robustly induced the levels of reactive oxygen species (ROS), as detected by staining with the fluorescent ROS indicator 2′,7′-dichlorodihydrofluorescein diacetate (DCFDA), which was significantly reduced by CQ (100 μM) and 4A7C-301 (1 μM) (Supplementary Fig. 1i). These effects were potentiated by Nurr1 OE but diminished by Nurr1 KD (Supplementary Fig. 1i). In addition, MPP⁺ treatment triggered a major mitochondrial dysfunction leading to reduced OXPHOS capacity (Supplementary Fig. 1j-o), which was efficiently restored by CQ (100 μM) and 4A7C-301 (1 μM), as shown by increases in basal respiration, maximal respiration, ATP turnover, as well as oxygen consumption rate (OCR) changes after FCCP injection, which effects were further enhanced by Nurr1 OE, but almost completely abrogated by Nurr1 KD (Supplementary Fig. 1j-o).

## 4A7C-301 restores Nurr1 protein expression levels that are diminished by exposure to environmental and genetic risk factors of PD in vitro

The above data showed that Nurr1 activation by its agonists can produce neuroprotective effects. To elucidate underlying molecular mechanisms, we next sought to address whether risk factors associated with PD (environmental and/or genetic) can alter the expression

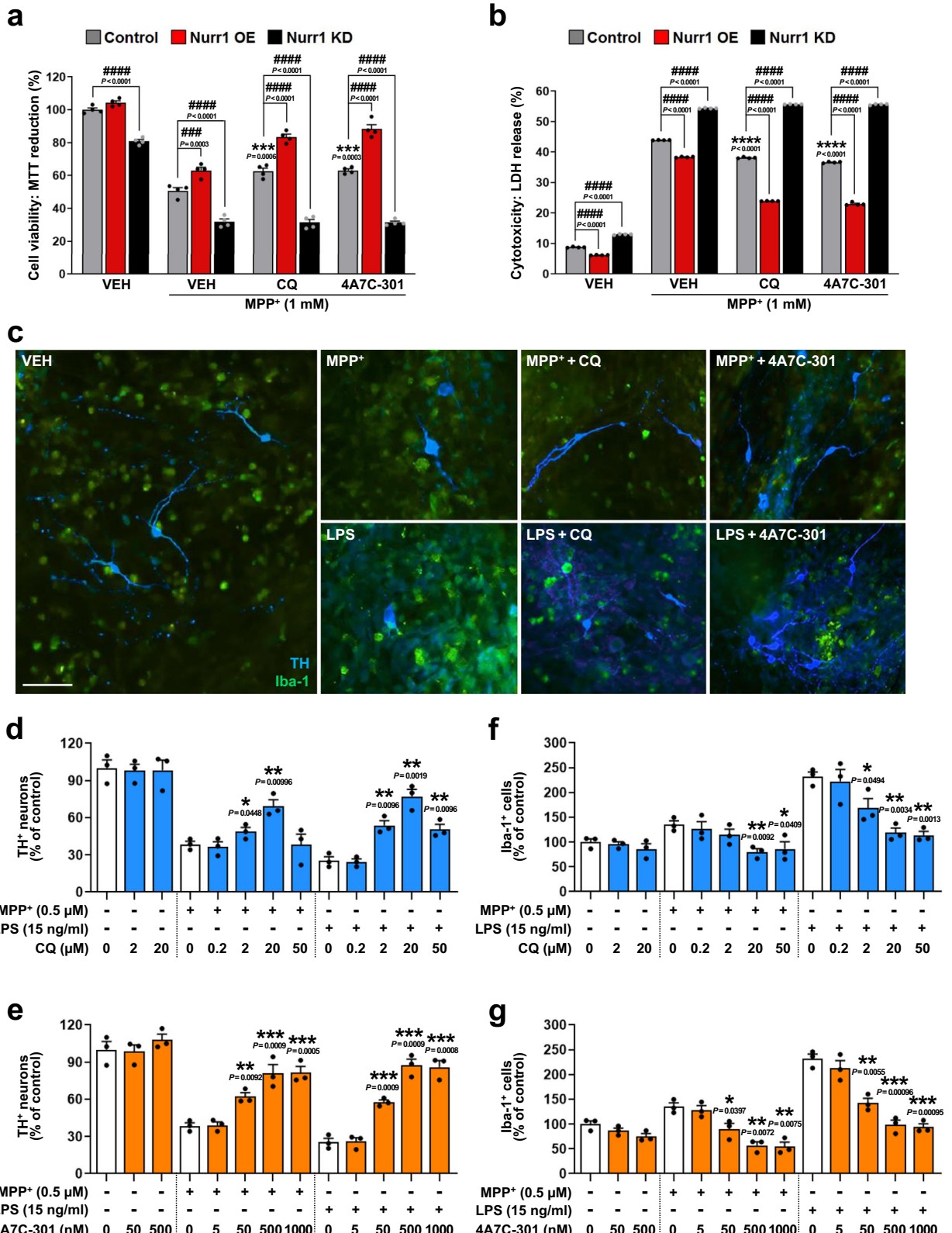

**Fig. 2 | In vitro comparison of CQ and 4A7C-301 neuroprotective function via Nurr1. a, b** Cell viability analyzed by MTT (3-(4,5-dimethylthiazol-2-yl)-2,5-diphe-nyltetrazolium bromide) reduction (**a**) and cytotoxicity measured by lactate dehydrogenase (LDH) release (**b**) with Nurr1 OE and KD in N27-A cells. ***$P < 0.001$, ****$P < 0.0001$ compared to vehicle (VEH) treatment under Control conditions; ###$P < 0.001$, ####$P < 0.0001$ compared between each treatment group, two-way ANOVA, Tukey's *post-hoc* test; $n = 4$ biologically independent samples per group. Data are mean ± s.e.m. **c** Representative TH (red) and Iba-1 (green)

immunofluorescence images from MPP+- or LPS-treated VM neuron-glia co-culture in the absence or presence of CQ or 4A7C-301. Scale bar, 100 μm.
**d–g** Quantification of TH+ neurons (**d**, **e**) and Iba-1+ microglia (**f**, **g**) counted and transformed as percentage of vehicle (VEH) control, from CQ (**d**, **f**) or 4A7C-301 (**e**, **g**) treated condition. *$P < 0.05$, **$P < 0.01$, ***$P < 0.001$ compared to 0 nM, mul-tiple unpaired two-tailed *t* test; $n = 3$ biologically independent samples per group. Data are mean ± s.e.m. from two biologically independent experiments.

of Nurr1 and other key transcription factors of mDANs. We first examined protein levels in MN9D cells after exposure to MPP$^+$ (0.5 mM). Remarkably, Nurr1 protein levels were significantly down-regulated, starting at 8 h, and decreased more than 70% following 24 h of exposure (Supplementary Fig. 4a, b). In sharp contrast, exposure to MPP$^+$ did not affect expression of other key factors (i.e., Pitx3, FoxA2, and Lmx1A) and β-actin protein. In addition, overexpression of wild-type (αSyn$^{WT}$) or mutant α-synuclein (αSyn$^{A53T}$) also significantly downregulated Nurr1 protein, but not other factors (Supplementary Fig. 4c–e). The mutant form (αSyn$^{A53T}$) exhibited greater effects than wild-type. These data are in line with previous studies reporting that Nurr1 expression is significantly reduced in PD patients[18–20,39] as well as in αSyn-overexpressing animal models of PD[40,41]. We also found that CQ and 4A7C-301 treatments significantly restored Nurr1 protein levels against MPP$^+$ and αSyn$^{WT}$ overexpression, but Nurr1 levels were maintained against αSyn$^{A53T}$ overexpression only by 4A7C-301 (Supplementary Fig. 4g, i). We next investigated whether CQ and 4A7C-301 regulate/increase *Nurr1* gene expression. Treatment with MPP$^+$ or overexpression of wild-type and mutant αSyn substantially reduced *Nurr1* mRNA expression in MN9D cells. However, treatment with CQ or 4A7C-301 did not alter *Nurr1* mRNA levels (Supplementary Fig. 4f, h) although protein levels were significantly restored (Supplementary Fig. 4g, i). Taken together, these data suggest that CQ and 4A7C-301 regulate the levels of Nurr1 protein at the post-transcriptional level without changing mRNA levels probably through enhancing Nurr1 protein's stability, which is vulnerable to and down-regulated by MPP$^+$ or αSyn toxicity. We further determined the effects of CQ and 4A7C-301 on dopaminergic gene expression including *TH*, dopamine transporter (*DAT*), aromatic L-amino acid decarboxylase (*AADC*), vesicular monoamine transporter 2 (*VMAT2*) and *c-Ret* in two in vitro dopaminergic systems, MN9D cell line and mouse ventral mesencephalic (VM) primary cells. CQ/4A7C-301 significantly rescued expression of all these genes tested that were decreased by MPP$^+$ or 6-OHDA treatment, and 4A7C-301 exhibited similar effects at 20–40 times lower concentrations than CQ (Supplementary Fig. 5). When Nurr1 was knocked down, not only did the basal expression of these genes fell more than 50%, but CQ and 4A7C-301's protective effects completely disappeared (Supplementary Fig. 5h, i).

### 4A7C-301 suppresses neuroinflammation in ventral mesence-phalic (VM) neuron-glia co-cultures

To further investigate the neuroprotective effects of CQ and 4A7C-301 in a more physiological context, we used a primary rat VM neuron-glia co-culture system, which allowed us to examine survival of mDANs as well as neuroinflammation by microglial activation. As shown in Fig. 2c–g, MPP$^+$ (0.5 μM) or LPS (15 ng/ml) treatment prominently induced mDAN cell death (>60%) together with activation of ionized calcium-binding adaptor molecule 1 (Iba-1)-positive microglia. Pre-treatment with CQ or 4A7C-301 significantly rescued mDANs in a dose-dependent manner. 4A7C-301 showed maximal effect at 500 nM, which is a 40 times lower concentration than required for a similar CQ effect (20 μM) (Fig. 2d, e). In addition, 4A7C-301 optimally suppressed microglial activation against LPS at 500 nM, which is 40 times lower than CQ (Fig. 2f, g). To address whether CQ and 4A7C-301 can suppress the expression of pro-inflammatory genes in microglia (in the absence of mDANs), we used LPS to induce inflammation in mouse microglia-derived BV2 cells, leading to a dramatic induction of the tumor-necrosis factor-α (*TNFα*) gene, which was robustly suppressed by CQ and 4A7C-301 (Supplementary Fig. 6). In this assay, 4A7C-301 exhibited a much higher potency than CQ, with EC$_{50}$ of ~1 and ~100 nM, respectively. Collectively, our data show that 4A7C-301 protects mDANs from MPP$^+$- and LPS-induced cell death while suppressing neuroinflammation with a much higher potency than CQ for both effects.

### 4A7C-301 restores autophagy against MPP$^+$ in vitro

CQ (and AQ) are known to inhibit autophagy by disrupting mature autophagosome-lysosome fusion to form autophagolysosomes (APL) at the late stage of the autophagy process[42–44]. Since dysregulated autophagy is implicated in PD pathology[7], we determined whether 4A7C-301 also affects autophagy. We first studied HeLa cells, which are widely used to study autophagy regulation. To this end, we employed the tandem mRFP-GFP-LC3 fluorescent assay in which GFP, but not mRFP, is degraded in acidic lysosomes[45]. Under starvation conditions with EBSS medium, autophagy induction was evidenced by increased numbers of both yellow (mRFP$^+$/GFP$^+$) and red (mRFP$^+$/GFP$^-$) dots, representing autophagosome and APL formation, respectively. As expected, incubation with a potent lysosomal proton pump inhibitor, bafilomycin A$_1$ (BafA$_1$), completely inhibited the final step of autophagy, as shown by increases of yellow dots, but not red dots, compared to the EBSS control condition (Supplementary Fig. 7a, b). CQ treatment significantly inhibited APL formation, as indicated by a reduced percentage of red dots among total LC3 dots (21.7%), compared to that of the EBSS condition (62.0%). In a sharp contrast, treatment with 4A7C-301 did not inhibit APL formation, as shown by 63.9% red dots among total LC3 dots. When we performed this assay in N27-A cells, we observed an identical pattern as in HeLa cells, showing that CQ, but not 4A7C-301, inhibits the autophagy process (Fig. 3a, b). Since CQ is known to affect lysosomal acidity[46], we examined its effect on lysosomal pH. BafA$_1$ and CQ, but not 4A7C-301, significantly increased lysosomal pH in both HeLa (Supplementary Fig. 7c, d) and N27-A cells (Fig. 3c, d), providing possible mechanisms underlying their differential effects on autophagy. We further analyzed the autophagic flux by Western blotting of stage-specific autophagy markers in HeLa cells. The expression changes of LC3B-II and p62 showed that autophagy was initiated but failed to terminate in the presence of BafA$_1$ and CQ (Supplementary Fig. 7e–g). In contrast, autophagy was completed in the presence of 4A7C-301, as evidenced by decreased p62 level (Supplementary Fig. 7g). We also analyzed the autophagic flux in N27-A cells with or without MPP$^+$ (1 mM) administration. As in HeLa cells, autophagy proceeded in the presence of 4A7C-301, but not of BafA$_1$ or CQ (Fig. 3e–g). When MPP$^+$ was administered, autophagy was disrupted, as shown by LC3B-II accumulation and consistent late-stage p62 levels. Intriguingly, 4A7C-301 treatment significantly restored autophagy in the presence of MPP$^+$ in N27-A cells, leading to successful p62 degradation (Fig. 3g). Finally, although 4A7C-301 exhibited its maximal effect at ≤1 μM in all assays, it did not inhibit autophagy even when 10-times higher concentrations were used (Supplementary Fig. 7h–j).

### 4A7C-301 improves motor (rotarod, pole, and cylinder tests) and non-motor (olfactory discrimination) behaviors in MPTP-induced male mice

The above findings prompted us to hypothesize that 4A7C-301 may exhibit potent disease-modifying effects in animal models of PD. To address this, we used two mouse models, one based on a neurotoxin (MPTP) and the other on a genetic factor (αSyn). First, we used a sub-chronic MPTP-induced (30 mg/kg/day, 5 days) male mouse model by administering CQ (40 mg/kg/day), 4A7C-301 (5 mg/kg/day), or L-DOPA (50 mg/kg/day) for 16 days (Fig. 4a). MPTP-treated mice manifested impaired motor function compared to the vehicle control group (VEH) in all three tests (rotarod, pole, and cylinder tests), which was significantly rescued by all three drugs with similar efficacy although 4A7C-301 was used at about one-tenth the concentrations of CQ and L-DOPA (Fig. 4b–d). Since olfactory dysfunction is a frequent prodrome of PD[47], we investigated whether these compounds could improve this non-motor dysfunction. In the olfactory discrimination test, MPTP-lesioned mice stayed less time in the old bedding indicating impairment in distinguishing between familiar and non-familiar odors. CQ and 4A7C-301, but not L-DOPA, significantly restored olfaction

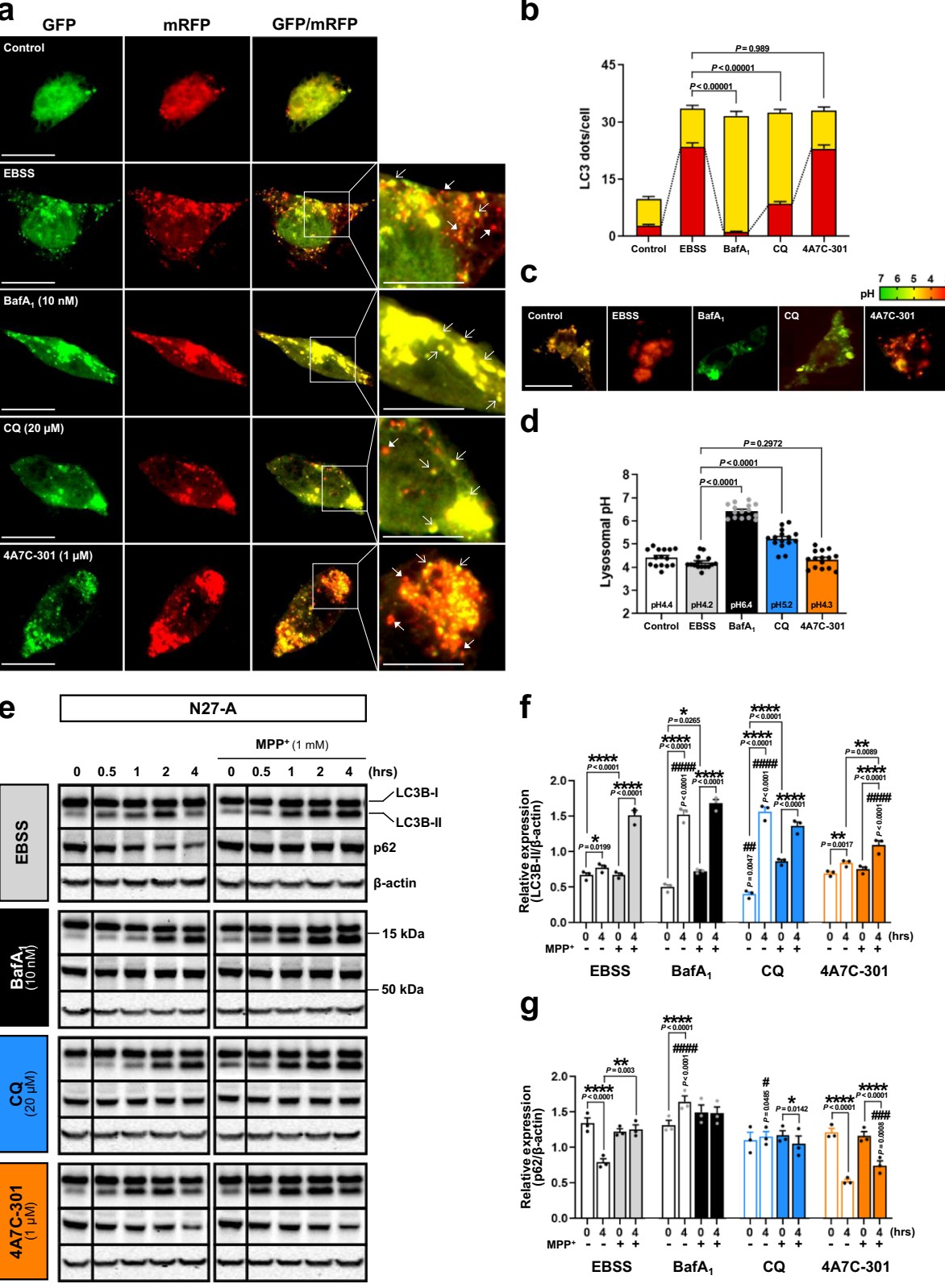

**Fig. 3 | CQ inhibits but 4A7C-301 protects autophagy. a**, **b** Autophagolysosome (APL) formation assay in N27-A cells. **a** Tandem mRFP-GFP-LC3 fluorescence images for APL detection. Scale bars, 20 μm. **b** Number of yellow LC3 dots and red LC3 dots per cell was counted from 10 random cells in each well from triplicates for each condition (total of 30 cells per each group). Two-tailed unpaired *t* test. Data are mean ± s.e.m. **c**, **d** Lysosomal pH detection in N27-A cells. **c** LysoSensor™ Yellow/Blue DND-160 fluorescence images. Scale bars, 20 μm. **d** Quantification from 5

random cells in each well from triplicates for each treatment group (total of 15 cells per each group). Two-tailed unpaired *t* test. Data are mean ± s.e.m. **e**–**g** Western blot analyses for autophagic flux in N27-A cells. Autophagic flux markers LC3B and p62 expressions (**e**) and quantitation of their expression levels (**f**, **g**). *$P < 0.05$, **$P < 0.01$, ****$P < 0.0001$; #$P < 0.05$, ##$P < 0.01$, ###$P < 0.001$, ####$P < 0.0001$ compared to EBSS, two-way ANOVA, Bonferroni's multiple comparisons; $n = 3$ biologically independent samples per group. Data are mean ± s.e.m.

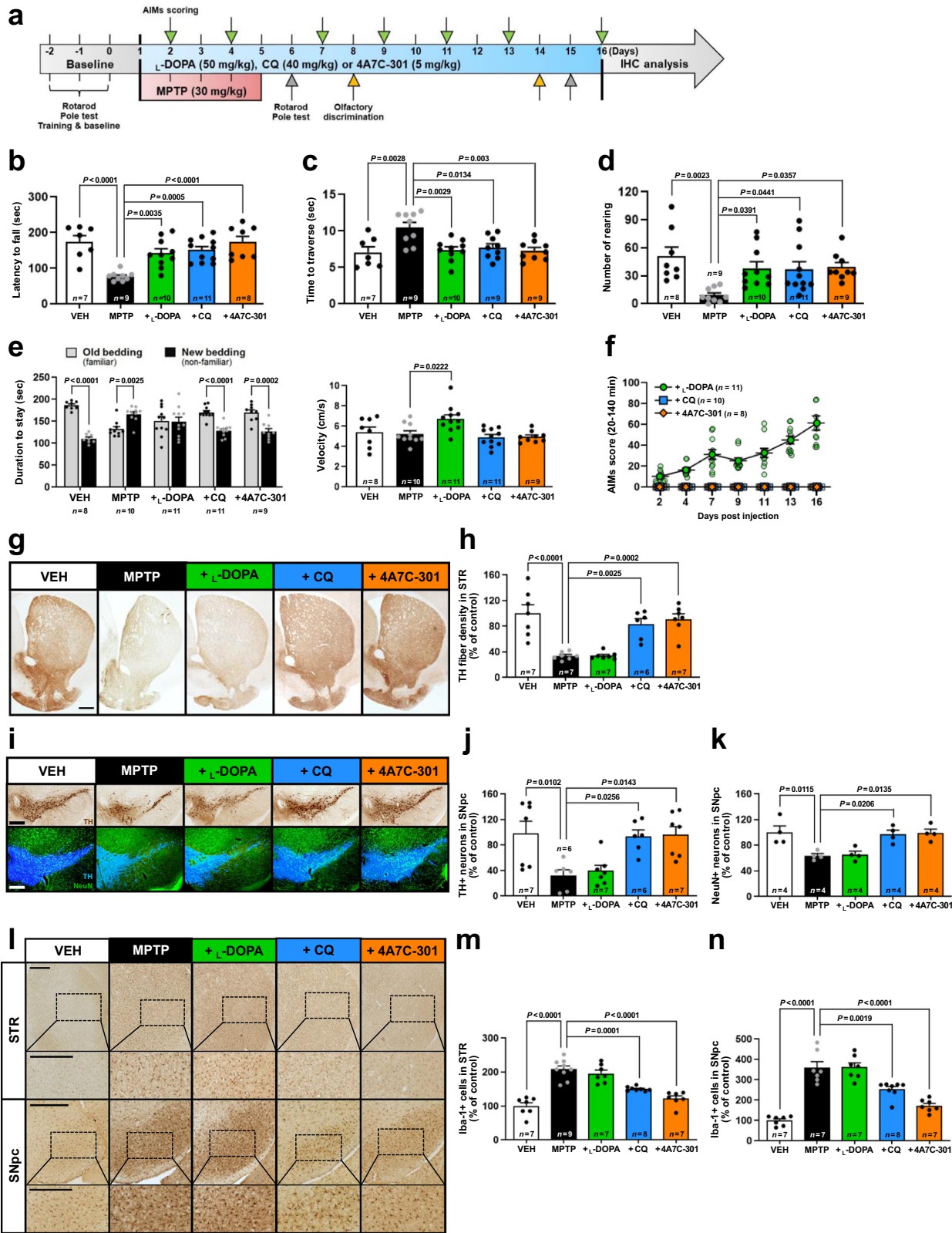

(Fig. 4e). Next, we examined abnormal involuntary movements (AIMs) scores including axial, limb, and orolingual dyskinesis, every other or third day, to monitor dyskinesia-like behavior. Concordant with the literature[48], L-DOPA treatment triggered severe AIMs from 7 days post injection (Fig. 4f and Supplementary Fig. 8f, g). In contrast, treatment with CQ or 4A7C-301 did not induce any detectable AIMs. To test

whether these behavioral improvements were due to neuroprotection, we stereologically analyzed immunohistochemical data and observed that TH+ fibers in the striatum (STR) and TH+ DA neurons and NeuN+ neurons in the substantia nigra (SN) were significantly retained after CQ and 4A7C-301 treatment but not by L-DOPA compared to the MPTP only group (Fig. 4g–k). We also examined whether neuroinflammation

**Fig. 4 | Neuroprotective effects of CQ and 4A7C-301 in MPTP-induced male mice. a** Schematic representation of L-DOPA, CQ, and 4A7C-301 administrations to MPTP-induced male mice. **b**–**d** Behavior tests involved in motor coordination and spontaneous movement assessed using rotarod (**b**), pole test (**c**), and cylinder test (**d**). One-way ANOVA, Tukey's *post-hoc* test. Data are mean ± s.e.m. **e** Behavior test involved in olfactory dysfunction assessed using olfactory discrimination test. Left, duration of stay between new and old bedding. Two-way ANOVA, Bonferroni's multiple comparisons. Data are mean ± s.e.m. Right, velocity

during the session. One-way ANOVA, Tukey's post-hoc test. Data are mean ± s.e.m. **f** Global AIMs score calculated by multiplying Basic and Amplitude AIMs scores (Supplementary Fig. 8f, g). Data are mean ± s.e.m. **g**–**k** Immunohistochemical analyses of TH+ DA neurons (**g**–**j**) and NeuN+ neuros (**l**, **k**) in the STR (**g**, **h**) and SNpc (**i**–**k**). Scale bars, 500 μm. One-way ANOVA, Tukey's *post-hoc* test. Data are mean ± s.e.m. **l**–**n** Immunohistochemical analyses of Iba-1+ microglia by counting in the STR (**m**) and SNpc (**n**). Scale bars, 500 μm. One-way ANOVA, Tukey's post-hoc test. Data are mean ± s.e.m.

was affected, using Iba-1 as an activated microglial marker. As shown in Fig. 4l–n, MPTP treatment significantly increased Iba-1+ microglia both in the STR and SN. Treatment with CQ and 4A7C-301, but not L-DOPA, markedly reduced Iba-1+ microglia in both regions. We further tested whether DA levels were restored by treatment with CQ and 4A7C-301. 4A7C-301 treatment significantly restored DA levels both in the SN and STR, whereas CQ did so only in the SN (Supplementary Fig. 10a–c). Finally, we repeated these behavioral tests at a chronic stage, at day 14–15, observing similar behavioral patterns in olfactory and AIMs tests (Supplementary Fig. 8b–e). However, only rotarod motor deficits remained, while in pole and cylinder tests deficits disappeared, suggesting some level of spontaneous recovery[49–51]. Taken together, the MPTP-lesioned mouse model demonstrated that 4A7C-301 improves both motor and non-motor deficits with decreased neuroinflammation and without any dyskinesia-like side effects. However, MPTP lesioning has limitations as a PD model including certain levels of spontaneous recovery. Thus, we next used the AAV2-αSyn-based mouse model, which more faithfully mimics the progressive and pathophysiological features related to human PD pathogenesis[52].

### 4A7C-301 rescues PD pathology in αSyn-induced male mice

We injected adeno-associated virus 2 (AAV2) expressing wild-type (αSyn^WT) or mutant αSyn (αSyn^A53T) unilaterally into the SN (left hemisphere) of male C57BL/6 J mice. CQ (40 mg/kg/day) or 4A7C-301 (5 mg/kg/day) was administered (i.p.) from the 4th to 8th week after surgery (Fig. 5a). Both αSyn^WT- and αSyn^A53T-induced mice manifested prominent loss of mDANs (Fig. 5b) and motor deficits in cylinder (Fig. 5c, d), rotarod (Supplementary Fig. 9a, c), and pole tests (Supplementary Fig. 9b, d) starting from the 6th week after surgery, with higher severity in αSyn^A53T-induced mice. CQ treatment led to partial recovery of rearing (Fig. 5c), latency to fall (Supplementary Fig. 9a), and time to traverse on the pole (Supplementary Fig. 9b). 4A7C-301 treatment improved motor deficits with greater potency than CQ in both αSyn^WT- and αSyn^A53T-induced mice on all tests assessed (Fig. 5d, Supplementary Fig. 9c, d). In the olfactory discrimination test, both αSyn^WT- and αSyn^A53T-induced mice exhibited olfactory dysfunction starting from the 4th week after surgery (Fig. 5e, f and Supplementary Fig. 9e–g). Both CQ and 4A7C-301 treatments rescued olfactory dysfunction in αSyn^WT-mice at 6 and 8 weeks after surgery. However, only 4A7C-301 significantly reversed olfactory dysfunction in αSyn^A53T-mice while there was no significant difference in velocity/mobility among the groups (Fig. 5f and Supplementary Fig. 9g). After behavioral tests, we sacrificed mice and characterized treatment effects on their brains by immunohistochemical analyses. Stereological quantification revealed that both CQ and 4A7C-301 treatments partially prevented the loss of TH+ and NeuN+ neurons and TH+ fibers in the SN and in the STR of αSyn^WT-mice, respectively (Fig. 5g–k). However, only 4A7C-301 treatment significantly prevented the loss of TH+ and NeuN+ neurons and fibers in αSyn^A53T-mice. In line with these data, both CQ and 4A7C-301 significantly increased DA levels in the SN and STR of αSyn^WT-mice while only 4A7C-301 increased DA levels of αSyn^A53T-mice (Supplementary Fig. 10d–f). We stereologically assessed immunoreactivity against phosphorylated αSyn at the serine 129 residue (αSyn^S129), which is associated with the facilitation of LB formation and accelerated neurodegeneration in PD[53]. As shown in Fig. 5l, m, αSyn^S129 was detected in the SN of αSyn^WT-mice, and more abundantly in αSyn^A53T-

mice. While CQ and 4A7C-301 treatments significantly diminished αSyn^S129 in αSyn^WT-mice, only 4A7C-301 significantly reduced it in αSyn^A53T-mice, suggesting that 4A7C-301 effectively alleviated pathogenic phenotypes in these animal models with a greater potency than CQ.

## Discussion

Despite numerous studies investigating PD pathophysiology, neither clear mechanisms underlying the loss of mDANs in the SN nor adequate disease-modifying treatments are yet available. Although it is well established that Nurr1 critically regulates the development, maintenance, and protection of mDANs, whether and how it is involved in PD pathogenesis is relatively undocumented. To address this issue, we tested whether environmental (e.g., MPP+) and/or genetic (e.g., αSyn) risk factors for PD compromise the expression and function of Nurr1. Our data revealed that both MPP+ treatment and αSyn overexpression robustly downregulate Nurr1 expression. In contrast, these treatments did not affect the expression of other key transcription factors of mDANs (e.g., Pitx3, FoxA2, and Lmx1A) or β-actin. Overexpression of pathogenic forms of αSyn resulted in greater reduction of Nurr1 levels than the wild-type. These data indicate that, though Nurr1 is critical for mDANs, as a gatekeeper, its continued exposure to PD risk factors (both genetic and environmental) compromises its expression and/or function, contributing to degeneration of mDANs. Since Nurr1's levels are downregulated in postmortem studies of aged brains[20], our model explains not only age-dependent loss of mDANs, but also environmental and genetic factor-induced loss. This conclusion is in line with previous human postmortem and animal model studies and further suggests the possibility that compromised expression/function of Nurr1 may underlie degeneration of mDANs in both familial and sporadic PD, providing a potential unifying mechanism of PD pathogenesis.

To address our hypothesis that Nurr1 is a promising molecular target for intervention in PD, we previously established high throughput screening systems to detect activators of Nurr1 and identified three compounds (i.e., CQ, AQ, and glafenine) that share an identical scaffold 4A7C. Based on these findings, we and others hypothesized that 4A7C represents a SAR for binding and activation of Nurr1[24,54]. Using this putative SAR (4A7C), we generated >570 derivatives of CQ and characterized them, resulting in the identification of an optimized agonist, 4A7C-301. Here, we tested 4A7C-301 and CQ for their effects on various in vitro and in vivo models of PD and propose that 4A7C-301 may provide a disease-modifying treatment for PD, as evidenced by the following results.

First, 4A7C-301 showed ~20-fold higher binding affinity compared to CQ or AQ and its activation of Nurr1's function requires the presence of specific amino acids within the Nurr1-LBD. Furthermore, 4A7C-301 robustly potentiated Nurr1's transcriptional activity and exhibited neuroprotective effects for mDANs exposed to neurotoxins (e.g., MPP+ and LPS) with greater potency than CQ (>20-fold), suggesting a mechanism-based Nurr1 activation. Our data revealed that 4A7C-301 effectively enhances both Nurr1's transcriptional activator function (e.g., expression of mDAN-specific genes and production of DA in mDANs) and its repressor function (e.g., suppression of pro-inflammatory cytokine genes and microglial activation) with equally greater efficiency than CQ (>20-fold). Second, we observed that

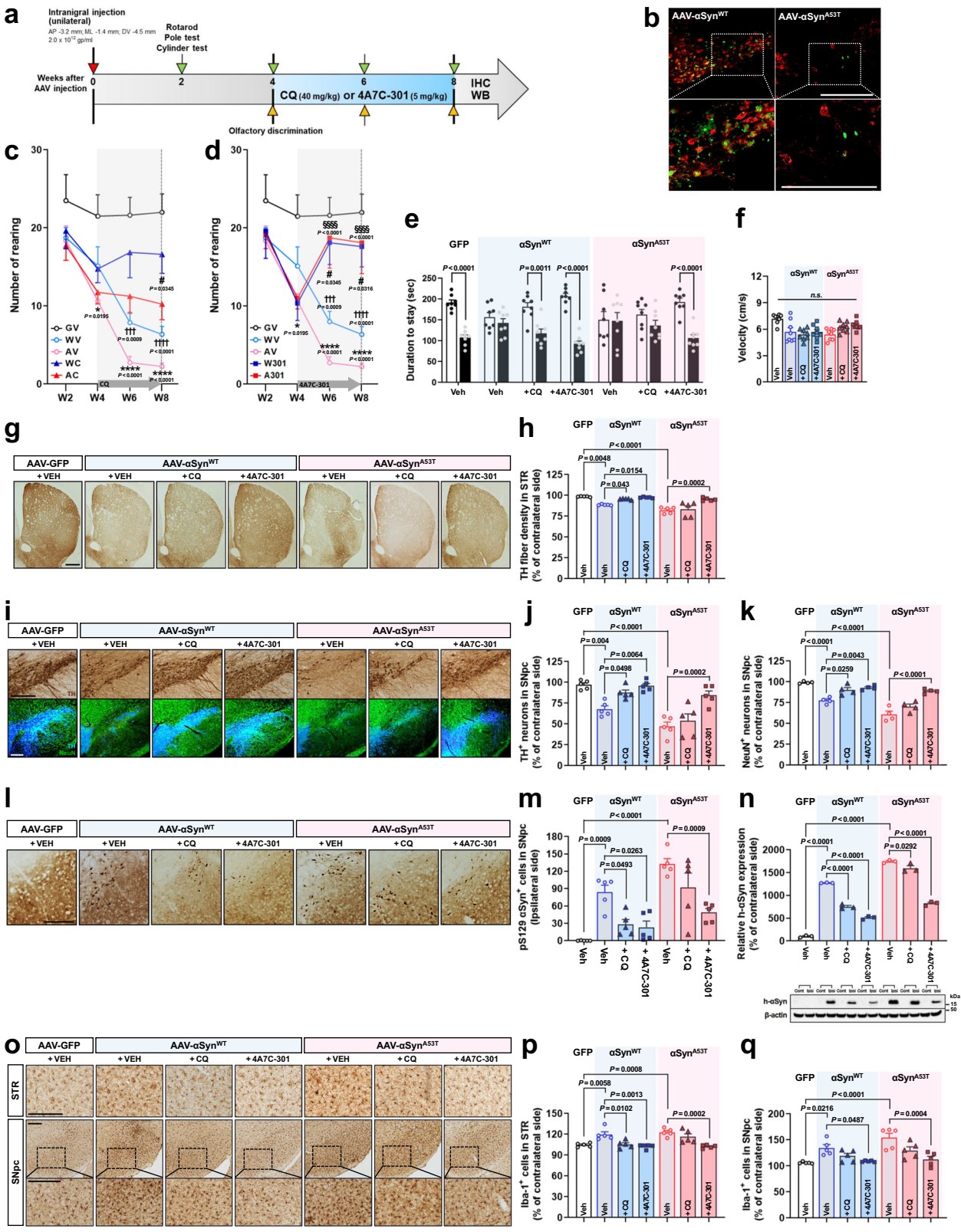

treatment with CQ or 4A7C-301 significantly restored Nurr1 protein levels that were reduced by MPP⁺ treatment and αSyn overexpression without change in their mRNA levels, strongly suggesting that Nurr1 ligands can regulate the levels of Nurr1 protein at the post-transcriptional levels. Thus, we speculate that Nurr1 ligands may regulate Nurr1 protein's stability, which is vulnerable and down-regulated by environmental (MPP⁺) and/or genetic (αSyn) toxicity and that

4A7C-301 (and CQ to a lesser degree) exhibits neuroprotective effects by two distinct mechanisms: (1) activating Nurr1's transcriptional function and (2) stabilizing and restoring Nurr1 protein levels. Third, using diverse cellular models, we found that 4A7C-301 substantially protects mDA cells via multiple downstream pathways such as reduction of oxidative stress and cytotoxicity, protection of mitochondrial function, induction of mDAN-specific gene expression, including

**Fig. 5 | 4A7C-301 exhibits robust neuroprotective effects on αSyn^WT- and αSyn^A53T-induced male mouse models. a** Schematic representation of CQ and 4A7C-301 administrations to AAV-αSyn-injected male mice. **b** Representative TH (red) and GFP (green) immunofluorescence images from two independent experiments in the SNpc of AAV-αSyn^WT- or -αSyn^A53T-injected mice. Scale bars, 500 μm. **c, d** Cylinder tests with CQ (**c**) or 4A7C-301 (**d**) treated group. $^{†††}P < 0.001$, $^{††††}P < 0.001$ compared between GV and WV; $^{*}P < 0.05$, $^{****}P < 0.0001$ compared between GV and AV; $^{#}P < 0.05$ compared WV to WC or W301; $^{§§§§}P < 0.0001$ compared AV to AC or A301. Two-way ANOVA, Tukey's multiple comparisons; $n = 8$ per group. Data are mean ± s.e.m. GV, GFP + VEH; WV, αSyn^WT + VEH; AV, αSyn^A53T + VEH; WC, αSyn^WT + CQ; AC, αSyn^A53T + CQ; W301, αSyn^WT + 4A7C-301; A301, αSyn^A53T + 4A7C-301. **e, f** Olfactory discrimination test performed at 8 weeks after surgery. **e** Duration of stay between new and old bedding. Two-way ANOVA, Bonferroni's multiple comparisons. Data are mean ± s.e.m. **f** Velocity during the session. *n.s.*, not significant ($P > 0.05$), one-way ANOVA. Data are mean ± s.e.m. $n = 8$ per group. **g, h** Immunohistochemical analyses of TH^+ neurons by densitometry in the STR. Scale bar, 500 μm. One-way ANOVA, Tukey's *post-hoc* test; $n = 5$ per group. Data are mean ± s.e.m. **i–k** Immunohistochemical analyses of TH^+ DA neurons (**i, j**) and NeuN^+ neuros (**i, k**) in the SNpc. Scale bars, 500 μm. One-way ANOVA, Tukey's post-hoc test; **j**, $n = 5$ per group; **k**, $n = 4$ per group. Data are mean ± s.e.m. **l** Representative pS129 immunohistochemistry in the SNpc. Scale bars, 250 μm. **m**, Quantification of pS129 αSyn^+ cells by counting. One-way ANOVA, Tukey's post-hoc test; $n = 5$ per group. Data are mean ± s.e.m. **n** Western blot analyses of human-αSyn in the SNpc of AAV-αSyn-injected mice. Values are expressed as a relative expression compared to contralateral side. One-way ANOVA, Tukey's post-hoc test; $n = 3$ per group. Data are mean ± s.e.m. **o–q** Immunohistochemical analyses of Iba-1^+ microglia by counting in the STR (**p**) and SNpc (**q**). One-way ANOVA, Tukey's *post-hoc* test; $n = 5$ per group. Data are mean ± s.e.m. Scale bars, 250 μm.

GDNF-receptor c-ret, and reduction of neuroinflammation. Furthermore, while CQ inhibits autophagy, 4A7C-301 does not. In fact, 4A7C-301 restores cellular autophagy functions impaired by MPP^+ treatment. Since CQ is a prototype autophagy inhibitor and known to accelerate the deposition of αSyn pathology[55], CQ's autophagy inhibition and 4A7C-301's autophagy protection is an important distinction. We speculate that the same core structure allows CQ and 4A7C-301 to bind Nurr1 (related to the putative SAR) and show similar activities in most assays, while their different side chain structures most likely underlie their distinct effects on autophagy. In support of this, CQ and BafA_1, both well-known autophagy inhibitors, significantly increased lysosomal pH, while 4A7C-301 had no effect, suggesting possible mechanisms underlying CQ/4A7C-301's different effects on autophagy. Fourth, in MPTP- and αSyn-induced mouse models, 4A7C-301 significantly restored motor deficits and non-motor behavior (e.g., olfactory defects) without any dyskinesia-like side effects. A major limitation of current pharmacological treatments of PD is that they eventually induce side effects such as dyskinesia in many patients[8,9]. As expected, L-DOPA administration induced robust dyskinesia-like behavior in MPTP-induced mice. In sharp contrast, despite comparable improvements in motor tests, treatment with 4A7C-301 or CQ did not induce any detectable dyskinesia-like behavior. A possible action mechanism of 4A7C-301/CQ in the MPTP model is that they may influence metabolism of MPTP to MPP^+. However, this explanation is unlikely because 4A7C-301/CQ were administered 6 h post MPTP injection, the time point when MPTP is completely metabolized into MPP^+ in both the striatum and SN[56].

Another interesting finding of this study is the observed olfactory deficit in both MPTP-induced and αSyn overexpression models, and that 4A7C-301 (and CQ to a lesser degree) significantly rescued this deficit, suggesting the possibility that Nurr1 agonists may ameliorate at least some of the non-motor deficits of PD as well. Recent studies using animal models with αSyn-overexpression[57] or αSyn preformed fibrils injection[58] also observed olfactory dysfunction. However, at present, the molecular mechanisms underlying olfactory dysfunction in these animal models and its amelioration by Nurr1 agonists remain elusive and await further investigation. Finally, our data revealed that Nurr1 and αSyn reciprocally regulate each other both in vitro and in vivo, which is prominently influenced by 4A7C-301. While overexpression of αSyn robustly downregulates Nurr1, conversely, accumulation of αSyn was significantly diminished by 4A7C-301 even when its pathogenic form (αSyn^S129) was overexpressed.

Despite these promising data, it must be noted that 4A7C-301 has limitations. For instance, the mode of action through which 4A7C-301 regulates Nurr1's transcriptional function and/or expression is unclear. This issue is critical because Nurr1 can function as both a transcriptional activator and a repressor in different cellular contexts and 4A7C-301 influences both functions. In addition, the potential off-target effects and toxicities of 4A7C-301 need to be fully addressed in assessing its therapeutic potential. In summary,

while further studies are warranted, we conclude that Nurr1 represents a promising target for the development of neuroprotective therapeutic agents. Optimized targeted agonists, such as 4A7C-301 or its analogous compounds, may have the potential as mechanism-based, disease-modifying treatments for both sporadic and familial PD.

## Methods
All experimental procedures followed the National Institute of Health guidelines and all protocols were approved by the Institutional Animal Care and Use Committee of McLean Hospital (Protocol #: 2015N000002).

### Synthetic procedure for 4A7C-301
All the chemicals used in the synthesis were purchased from Sigma-Aldrich and were used without further modifications. Thin layer chromatography was used to monitor the progress of the reactions using pre-coated TLC plates (E. Merck Kieselgel 60 F254) with spots being visualized by iodine vapors/UV lamp. Compounds were purified over silica gel (60-120 mesh) column or recrystallized with suitable solvents. Solvents were distilled before use for purification purposes. Melting points were recorded on an ERS automated melting point apparatus and are uncorrected. IR spectra were recorded using a Perkin-Elmer and a Bruker FT-IR and the values are expressed as $\lambda_{max}$ cm^{-1}. Mass spectral data were recorded on a Jeol-AccuTOF JMS-T100LC and micromass LCT Mass Spectrometer/Data system. The ^1H NMR and ^13C NMR spectra were recorded on a Jeol Spectrospin spectrometer at 400 MHz and 100 MHz, respectively, using TMS as an internal standard. The chemical shift values are recorded on δ scale, and the coupling constants (*J*) are in Hz.

**Procedure for the synthesis of intermediate 3.** A mixture of 4,7-dichloroquinoline (**1**, 10.0 g, 50.49 mmol) and ethane-1,2-diamine (**2a**, 16.9 mL, 252.45 mmol) was stirred at 120 °C for 6 h (Scheme 1). The reaction mixture was then cooled down to room temperature and ice-cold water was added to it. The solid thus obtained was filtered and washed with excess water. The crude product was crystallized by using ethanol and was used as such for the next step.

**Procedure for the synthesis of intermediates 5 A and 5B.** Intermediate 3 (5 g, 22.6 mmol) and compound 4 (4.14 g, 22.6 mmol) were taken in round bottom flask containing 50 mL THF. Triethylamine (9.5 mL, 67.8 mmol) was added drop-wise into the above reaction mixture. After completion of addition, the reaction mixture was stirred overnight at 60 °C. After completion of the reaction, as observed by TLC, the reaction mixture was poured into crushed ice to precipitate the mixture of crude compounds 5 A and 5B. The crude product was purified by column chromatography using 5% MeOH–CHCl_3 as the eluent to afford **5 A** (25% yield, minor) and **5B** (53% yield, major) as the two pyrimidine regio-isomers 5 A and 5B.

$N^1$-(7-Chloroquinolin-4-yl)-$N^2$-(4,6-dichloropyrimidin-2-yl)ethane-1,2-diamine (5 A). White solid, Yield: 25%; mp: 237-239 °C; IR (KBr, cm⁻¹): 3262, 3113, 1549, 1445, 1234, 1128, 1097, 806; ¹H NMR (400 MHz, DMSO-$d_6$): $\delta$ (ppm) 3.45-3.53 (m, 4H), 6.71 (d, $J = 5.50$ Hz, 1H), 6.90 (s, 1H), 7.39-7.46 (m, 2H), 7.79 (d, $J = 2.20$ Hz, 1H), 8.19 (d, $J = 9.07$ Hz, 1H), 8.27 (d, $J = 5.77$ Hz, 1H), 8.41 (d, $J = 5.36$ Hz, 1H); anal. calcd for $C_{15}H_{12}Cl_3N_5$: C, 48.87; H, 3.28; N, 19.00; found: C, 48.96; H, 3.30; N, 18.97; ESI-HRMS (m/z) calcd for $C_{15}H_{13}Cl_3N_5$ (M + H)⁺: 368.0231, found: 368.0237 (M + H)⁺, 370.0211 (MH + 2)⁺, 372.0178 (MH + 4)⁺.

$N^1$-(7-Chloroquinolin-4-yl)-$N^2$-(2,6-dichloropyrimidin-4-yl)ethane-1,2-diamine (5B). White solid, Yield: 53%; mp: 235-237 °C; IR (KBr, cm⁻¹): 3348, 2940, 1566, 1447, 1358, 1271, 1120, 984, 804; ¹H NMR (400 MHz, DMSO-$d_6$): $\delta$ (ppm) 3.45-3.60 (m, 4H), 6.54 (s, 1H), 6.70 (d, $J = 5.36$ Hz, 1H), 7.44-7.47 (m, 2H), 7.79 (d, $J = 1.92$ Hz, 1H), 8.20 (d, $J = 9.07$ Hz, 1H), 8.37-8.43 (m, 2H); anal. calcd for $C_{15}H_{12}Cl_3N_5$: C, 48.87; H, 3.28; N, 19.00; found: C, 49.01; H, 3.30; N, 19.05; ESI-HRMS (m/z) calcd for $C_{15}H_{13}Cl_3N_5$ (M + H)⁺: 368.0231, found: 368.0242 (M + H)⁺, 370.0214 (MH + 2)⁺, 372.0181 (MH + 4)⁺.

**Procedure for the synthesis of compound N¹-(4,6-bis(4-ethylpiperazin-1-yl)pyrimidin-2-yl)-N²-(7-chloroquinolin-4-yl)ethane-1,2-diamine (4A7C-301).** Compound 5 A (200 mg, 0.54 mmol) and N-ethylpiperazine (1.4 mL, 10.85 mmol) were added to a sealed tube and the reaction mixture was heated for 5 hours at 110 °C. After completion of the reaction, the mixture was poured into ice cold water. The resulting precipitate was filtered, washed with excess cold water to remove the amine, and dried to get the crude product. It was then recrystallized in ethanol to get the pure product as an off-white solid. Yield: 70%; mp: 88-90 °C; IR (KBr, cm⁻¹): 3282, 2924, 2852, 1569, 1446, 1372, 1260, 1006, 798; ¹H NMR (400 MHz, CDCl₃): $\delta$ (ppm) 1.09 (t, $J = 7.33$ Hz, 6H), 2.40-2.51 (m, 12H), 3.38-3.39 (m, 2H), 3.60 (brs, 8H), 3.83-3.88 (m, 2H), 5.05 (t, $J = 6.18$ Hz, 1H), 5.22 (s, 1H), 6.26 (d, $J = 5.50$ Hz, 1H), 7.21 (dd, $J = 8.93$, 2.29 Hz, 1H), 7.31 (brs, 1H), 7.53 (d, $J = 8.70$ Hz, 1H), 7.88 (d, $J = 1.83$ Hz, 1H), 8.46 (d, $J = 5.50$ Hz, 1H); ¹³C NMR (100 MHz, CDCl₃): $\delta$ (ppm) 11.98, 39.62, 44.53, 47.43, 52.46, 52.59, 74.10, 98.12, 117.36, 122.61, 125.26, 128.27, 134.58, 149.04, 150.66, 152.05, 162.86, 164.70; anal. calcd for $C_{27}H_{38}ClN_9$: C, 61.87; H, 7.31; N, 24.05; found: C, 62.01; H, 7.34; N, 23.99; ESI-HRMS (m/z) calcd for $C_{27}H_{39}ClN_9$ (M + H)⁺: 524.3017, found: 524.3019 (M + H)⁺, 526.2991 (MH + 2)⁺; HPLC purity: 97.75.

## Cell culture

SK-N-BE(2)C human neuroblastoma, BV2 mouse microglia (ATCC), HeLa and MN9D cell lines were cultured in Dulbecco's Modified Eagle Medium (DMEM) supplemented with 10% fetal bovine serum (FBS) and 100 U/ml penicillin and 100 μg/ml streptomycin. The murine MN9D dopaminergic cell line was kindly provided by Drs. Michael Zigmond and Juliann Jaumotte from the University of Pittsburgh. The N27-A rat dopaminergic cell line was kindly provided by Dr. Curt Freed from the University of Colorado School of Medicine. N27-A cells were cultured in RPMI 1640 medium (Lonza) supplemented with 10% FBS and 1% penicillin/streptomycin[36]. For transactivity assay, SK-N-BE(2)C and N27-A cell lines were transfected with pCMV-fNurr1, pGAL-Nurr1(LBD), pGAL-Nor1(LBD) or pGAL-Nur77(LBD) and reporter construct (p4xNL3-Luc for fNurr1 or p8xUAS-Luc for LBD constructs). pRSV-β-gal was co-transfected as an internal control[23]. Cells were lysed in a lysis buffer (25 mM Tris-phosphate (pH 7.8), 2 mM DTT, 2 mM DCTA (1,2-diaminocyclohexane-N,N,N',N'-tetraacetic acid), 10% glycerol, and 1% Triton X-100) 24 h after transfection, followed by adding firefly luciferase substrate for luminescence detection measured by a luminometer plate reader (SpectraMax®; Molecular device). Nurr1 transactivity was normalized to β-galactosidase activity. For MTT and LDH assays, N27-A cells were treated overnight with 1-methyl-4-phenylpyridinium (MPP⁺, 1 mM) in the absence or presence of CQ or

4A7C-301. For Nurr1 stability determination, MN9D cells were incubated with MPP⁺ (0.5 mM) for 2, 4, 8, 16, 24, and 32 h, or were transduced with lentivirus expressing GFP only or αSyn (wild-type or A53T) then harvested 24, 48 and 72 h after transduction. For autophagolysosome (APL) formation assay, HeLa and N27-A cells were transfected with mRFP-GFP-LC3 (Addgene) using Lipofectamine 2000 (Invitrogen). The medium was changed with starvation medium (Earle's Balanced Salt Solution; EBSS) 24 h after transfection for autophagy induction.

## Western blot

Protein samples were prepared from cell harvests or brain tissues in homogenization buffer (50 mM Tris-HCl (pH 8.0), 150 mM NaCl, 0.5 mM EDTA, 1% Triton X-100 (Sigma-Aldrich) containing 1 mM PMSF, protease inhibitor cocktail (Roche) and phosphatase inhibitor cocktail (Sigma-Aldrich). Equal amounts of total protein determined by the Bradford method were loaded on a Bolt™ 4-12% Bis-Tris Plus gel (Invitrogen), followed by transfer to PVDF membranes (Bio-Rad). After blocking in TBS-T buffer (20 mM Tris-HCl (pH 7.4), 500 mM NaCl, 1% Tween-20) containing 5% skim milk, membranes were incubated with the following primary antibodies: Nurr1 (1:1,000; prepared in our laboratory[59]), LC3B (1:1,000; Cell Signaling, #2775), SQSTM1/p62 (1:1,000; Abcam, ab155686), αSyn (1:1,000; GeneTex, GTX112799), Pitx3 (1:1,000; Invitrogen, 38-2850), FoxA2 (1:1,000; Abnova, H00003170-M12), Lmx1A (1:1,000; Millipore, AB10533) and β-actin (1:5,000; Abcam, ab8227). Experiments were confirmed by duplicate measurements of the same sample. Quantification of immunoreactive bands was performed using ImageJ software (NIH, Bethesda, MD, USA) and expressed as a relative ratio against β-actin.

## RNA isolation and Quantitative Real-Time Polymerase Chain Reaction (qRT-PCR)

Total RNA was extracted using the GeneJET RNA Purification Kit (Thermo Fisher Scientific) and 100 ng of RNA was subjected to cDNA synthesis using Invitrogen Superscript cDNA Synthesis Kit (Invitrogen) according to the manufacturer's protocol. qRT-PCR was performed using a CFX Connect Real-Time System (Bio-Rad) in a 20 μl volume mixture containing 2X SsoAdvanced Universal SYBR Green Supermix (Bio-Rad), 100 ng/μl of forward and reverse primers, and 1 μl of cDNA sample. Gene expression levels were quantified by the $2^{\Delta\Delta CT}$ method ($C_t$ of target gene−$C_t$ of glyceraldehyde-3-phosphate dehydrogenase (GAPDH). The following primers were used for qRT-PCR: *TH* forward: 5'-CAAGGTTCCCTGGTTCCCAA-3', reverse: 5'-CTTCAGCGTGGCGTATACCT-3'; *DAT* forward: 5'-GTCACCAACGGTGGCATCTA-3', reverse: 5'-TAGGCTCCATAGTGTGGGGG-3'; *AADC* forward: 5'-CACGGCTAGCTCATACCCAG-3', reverse: 5'-GCTCTTCCAGCCAAAAAGGC-3'; *VMAT2* forward: 5'-ATGTGTTCCCGAAAGTGGCA-3', reverse: 5'-AAGTTGGGGAGCGATGAGTCC-3'; *c-Ret* forward: 5'-TGCTGCTCTGGGAGATTGTG-3', reverse: 5'-AACACTGGCCTCTTGTCTGG-3'; *TNFα* forward: 5'-ATAGCTCCCAGAAAAGCAAGC-3'; reverse: 5'-CACCCCGAAGTTCAGTAGACA-3'; *GAPDH*, forward 5'-GAAGGTCGGTGTGAACGGAT-3', reverse 5'-TTCCCATTCTCGGCCTTGAC-3'.

## Site-directed mutagenesis

Nurr1 mutants were generated using the QuikChange ‖ XL site-directed mutagenesis kit (Agilent Technologies) according to the manufacturer's instruction. Mutagenesis reactions were carried out on the plasmid of mouse Nurr1 cloned into pcDNA3.1 backbone, and then verified by sequencing.

## Binding and competition assays

Saturation and competition assays were performed as previously described[33]. For saturation assay, 0.2 μM of Nurr1-LBD and 125-10,000 nM of [³H]-CQ were incubated in binding buffer (20 mM sodium acetate buffer (pH 5.2)) at 4 °C overnight. For competition

assay, a fixed concentration of $[^3H]$-CQ (1,000 nM) was incubated with serial concentrations of unlabeled competitors (CQ, 4A7C-301 or retinoic acid (RA)) ($10^{-4}$–$10^5$ nM) in binding buffer at 4 °C overnight. Assay mixtures were applied to GF/B filters (Brandel Inc.) to capture Nurr1-LBD bounded ligands. The data indicate the average of triplicates, and the competition assay values were transformed as a percentage of inhibition relative to 0 nM of competitors. All graphs and $IC_{50}$ values were generated using the non-linear regression program in GraphPad Prism version 8.0.2.

## Time-resolved fluorescence resonance energy transfer (TR-FRET) assay

TR-FRET assay was performed in PPI Terbium (Tb) detection buffer (PerkinElmer) and contained 0.2 μM of Nurr1-LBD, 0.5 μM of fluorescence-labeled hydroxy-CQ (HCQFluo; Proimaging) and serial concentrations of CQ, 4A7C-301 or RA ($10^{-4}$–$10^6$ nM). Rabbit anti-Nurr1 antibody (3 nM; prepared in our lab) and anti-rabbit IgG-Tb (1:400; Cisbio, 61PARTAF) were included in the assay solution as a donor complex. Assay solutions were loaded in a 384-well OptiPlate™ (PerkinElmer). After 2 h incubation in the dark at room temperature, the FRET signal was measured by excitation at 340 nm and emission at 520 nm for fluorescein and 495 nm for Tb using a VICTOR Nivo plate reader (PerkinElmer). Data were fit using the non-linear regression program in GraphPad Prism version 8.0.2.

## Nurr1 overexpression and knockdown

Cells were transfected with Nurr1 subcloned into pcDNA3.1 vector[23] for Nurr1 overexpression (OE). For Nurr1 knockdown (KD), specific Nurr1 shRNA (V3LHS_411033, TCTTCTGAACAACAAACTG; GE Dharmacon) was selected and transfected. pcDNA and scrambled shRNA (#RHS4346) were used for negative control for Nurr1 OE or KD, respectively. Nurr1 OE or KD by transfection was analyzed by Western blot.

## Lentivirus production and transduction

Lenti-GFP, -αSyn[WT], and -αSyn[A53T] viruses were generated separately by co-transfecting each shuttle plasmid with lentivirus packaging plasmids into HEK293T cells using lipofectamine 2000 (Invitrogen). Briefly, 2 μg each FUW inducible shuttle plasmid, 2 μg FUW-rtTA, 2 μg psPax2 and 1.5 μg MD2.G were mixed and transfected into HEK293T cells. After 48 h, viral supernatants were collected and centrifuged at 1000$g$ for 10 min, and subsequently filtered through a 0.45 μm filter. Lentiviruses were transduced into MN9D cells with 8 μg/ml of polybrene and the viral supernatant was removed and replaced with fresh medium a day after transduction.

## Cellular oxidative stress, cytotoxicity, and cell viability assays

Cellular oxidative stress was detected using the DCFDA Cellular ROS Assay Kit (Abcam) according to the manufacturer's protocol. Briefly, cells seeded in black clear bottom 96 well plate were stained with DCFDA solution (20 μM) for 45 min at 37 °C. Fluorescence was measured at Ex/Em = 485/535 nm using a microplate reader (Synergy HT; Bio-Tek) and ROS levels were calculated as relative to vehicle control. Cytotoxicity was determined by measuring lactate dehydrogenase (LDH) release into the culture medium using a LDH cytotoxicity detection kit (Roche)[33]. Absorbance was measured at 490 nm with a microplate reader and cytotoxicity was calculated as a percentage (%) relative to 1% TritonX-100 control. Cell viability was determined using 3-(4,5-dimethylthiazol-2-yl)-2,5-diphenyltetrazolium bromide (MTT) as previously described[60]. Briefly, cells were incubated with 5 mg/ml of MTT (Sigma-Aldrich) solution for 3.5 h at 37 °C. MTT formazan crystals were dissolved in 10 times the volume of MTT solvent (4 mM HCl and 0.1% Nonidet P-40 in isopropanol) and the absorbance from the generated blue formazan was measured at 570 nm after 15 min incubation

at room temperature. Cell viability was calculated as a percentage (%) of MTT reduction relative to vehicle control.

## Mitochondrial activity assay

Mitochondrial activity was measured using the Seahorse XFp analyzer (Agilent Technologies) according to the manufacturer's instruction. Briefly, cells were plated onto wells of a XFp cell culture miniplate and incubated in a 37 °C incubator overnight. The assay was performed after cells were equilibrated for 1 h in XF assay medium supplemented with 10 mM D-glucose, 5 mM sodium pyruvate, and 2 mM glutamax in a non-$CO_2$ incubator. Mitochondrial activity was monitored through sequential injections of 1 μM oligomycin, 2 μM carbonyl cyanide-4-(trifluoromethoxy) phenylhydrazone (FCCP) and 0.5 μM rotenone/antimycin A to calculate basal respiration (= baseline oxygen consumption rate (OCR)−rotenone/antimycin A OCR), maximal respiration (= FCCP OCR−rotenone/antimycin A OCR), ATP turnover (= baseline OCR−oligomycin OCR), and OCR changes after FCCP injection (= FCCP OCR−oligomycin OCR). Each plotted value was normalized to total protein quantified using a Bradford protein assay (Bio-Rad) and analyzed using the Seahorse WAVE Desktop software (Agilent Technologies).

## Primary midbrain dopaminergic neuronal culture

Primary ventral mesencephalic (VM) dopaminergic neurons were obtained from embryonic day 13.5 (E13.5) mice embryos (C57BL/6 J; Jackson Laboratory). Briefly, dissected VM brain tissues were collected in ice-cold Dulbecco's phosphate-buffered saline (DPBS) and then mechanically dissociated using pipettes. Thereafter, dissociated tissues were incubated in 0.05 % Trypsin-EDTA (Invitrogen) for 5 min at 37 °C. The cell suspension was plated onto glass coverslips in 48-well plates pre-coated with 5 mg/ml poly-L-ornithine, 5 μg/ml Fibronectin or 1 μg/ml Laminin at a density of $2.0 \times 10^5$ cells/$cm^2$ in differentiation medium (1% heat-inactivated FBS, 2 mM L-glutamine, 50X B27 supplement, 100 μM ascorbic acid and 2 μg/ml bFGF in Neurobasal media). Primary VM neurons were treated with CQ (20 μM) or 4A7C-301 (0.5 μM) with or without 6-OHDA (20 μM) and harvested to determine dopaminergic gene expression by real-time PCR.

## Primary rat ventral mesencephalic (VM) neuron-glia co-culture and immunocytochemistry (ICC)

Primary rat VM neuron-glia co-culture was prepared from E14 rat embryos (Fisher 344; Charles River)[33]. On day 7, cells were pre-treated with CQ or 4A7C-301 30 min prior to MPP+ (0.5 μM) or lipopolysaccharide (LPS, 15 ng/ml) treatment and then fixed in 4% formaldehyde (Sigma-Aldrich) for ICC 7 days after treatments (14 days in vitro). After blocking with 10% normal donkey serum (NDS) and 0.1% Triton X-100 solution for 1 h at room temperature, cells were incubated with primary antibodies against TH (1:1,000; Pel Freez, P40401) and Iba-1 (1:500; Abcam, ab5076) in 1% NDS and 0.1% Triton X-100 solution for overnight at 4 °C. Alexa Fluor 568- or Alexa Fluor 488-conjugated anti-rabbit or anti-goat secondary antibodies (1:500; Invitrogen) were incubated following the primary antibody incubation, and then cover glasses were mounted using FluoShield™ mounting media (Sigma-Aldrich). Fluorescence images were taken using a KEYENCE microscope (BZ-X800; KEYENCE).

## Autophagy assays

**Autophagolysosome formation.** HeLa and N27-A cells were transfected with tandem mRFP-GFP-LC3 (Addgene) plasmid using Lipofectamine 2000 (Invitrogen) in 24-well plates. After 24 h, cells were incubated in starvation medium (Earle's Balanced Salt Solution, EBSS; Lonza) containing bafilomycin A1 (BafA1, 10 nM), CQ (20 μM), or 4A7C-301 (1 μM) for 6 h and then fixed with 4% formaldehyde for fluorescence detection. Images were taken using a KEYENCE microscope and

red (mRFP) LC3 dots and yellow (mRFP+GFP) LC3 dots were counted from three independent wells for each treatment group. 10 cells were randomly selected and counted from each well and a total of 30 cells were analyzed for each treatment group in both HeLa and N27-A cells.

**Lysosomal pH detection.** Lysosomal pH was detected as previously described[61]. Briefly, after incubation in EBSS containing BafA1 (10 nM), CQ (20 μM), or 4A7C-301 (1 μM) for 6 h, HeLa and N27-A cells were treated with LysoSensor™ Yellow/Blue DND-160 (5 μM; Life Technology) for 45 min, followed by 2 washes in HEPES buffer (Gibco). Fluorescence images at excitation/emission wavelengths of 360/460 nm and 470/525 nm were taken immediately using a 400 nm dichroic mirror (KEYENCE). The ratio of fluorescence intensities ($F_{360/470}$) were measured and calculated manually using ImageJ for each cell. The average pH from measuring 5 cells from each of three wells for each treatment group in both HeLa and N27-A cells was calculated by fitting the $F_{360/470}$ values to a plotted pH calibration curve with KCl/NaCl buffered from pH 3.5 to 7.0 using GraphPad Prism (Version 8.0.2).

**Autophagic flux.** Autophagic flux was determined by Western blot against LC3B (1:1,000; Cell Signaling, #2775) or SQSTM1/p62 (1:1,000; Abcam, ab155686) from cells harvested 0.5, 1, 2, and 4 h after autophagy induction. The band intensities were quantified using ImageJ and normalized against β-actin.

## Sub-chronic MPTP-Induced PD model mice

Male C57BL/6 J mice (8–10 weeks, 25–30 g; Jackson Laboratories) were randomly assigned to 5 groups including vehicle (VEH), MPTP, MPTP + L-DOPA, MPTP + CQ and MPTP + 4A7C-301 ($n \geq 7$ for each group). MPTP (Sigma-Aldrich) was intraperitoneally (i.p.) injected (30 mg/kg) for 5 days (Day 1-5). Different dosages of CQ (20, 40, and 80 mg/kg) and 4A7C-301 (1, 5, 20 mg/kg) were first tested to determine the effective dose window, then 40 and 5 mg/kg doses were finally selected for CQ and 4A7C-301 respectively to show the most prominent effects. CQ or 4A7C-301 was dissolved in PBS and administrated (i.p.) for 16 days starting with MPTP injection (Day 1-16). To test dyskinetic involuntary behavior, an L-DOPA injected group was included, which received 50 mg/kg of L-DOPA and 15 mg/kg of benserazide (Sigma-Aldrich) (i.p.) for 16 days (Day 1–16). All compounds were administered 6 h post-MPTP injection to avoid any effect of compounds on the MPTP metabolism in the brain. Motor and non-motor related behavior tests were performed on the following days as described in Fig. 4a in an assessor- and observer-blinded fashion. Pre-training and baseline setup for motor-related behaviors were performed 3 days prior to the injections. Mice were sacrificed and subjected to immunohistochemistry. Mice were housed at the Animal Care Facilities of McLean Hospital under a 12-h light/12-h dark cycle, ambient temperature and humidity. Animals were handled in accordance with McLean Hospital's Institutional Animal Care and Use Committee (Protocol #: 2015N000002) and followed National Institutes of Health guidelines.

## AAV-αSyn-induced PD model mice

Adeno-associated viruses (AAVs) expressing GFP only or αSyn (wild-type or A53T) were kindly provided by Dr. Yoon Seong Kim from the University of Central Florida. Recombinant AAV vectors were produced using modified pAAV-IRES-hrGFP (Agilent) to express wild-type and mutant human αSyn under control of the CMV promoter. Viral particles (AAV2) were produced at the University of Iowa Viral Vector Core according to standard operating procedures (https://medicine.uiowa.edu/vectorcore/). Male C57BL/6 mice (8-10 weeks, 25-30 g; Jackson Laboratories) were randomly assigned to 7 groups including empty vector (GFP) + VEH, wild-type αSyn (αSyn$^{WT}$) + VEH, mutant αSyn (αSyn$^{A53T}$) + VEH, αSyn$^{WT}$ + CQ, αSyn$^{A53T}$ + CQ, αSyn$^{WT}$ + 4A7C-301

and αSyn$^{A53T}$ + 4A7C-301 ($n = 8$ per group). AAV vectors were delivered into the substantia nigra unilaterally by stereotaxic surgery. Mice were anesthetized using a gaseous mixture of oxygen and isoflurane (2.5% induction, 1.8% maintenance), and the AAV vectors were delivered into the left hemisphere (A/P -3.2 mm; M/L -1.4 mm; D/V -4.5 mm) as a single 2 μl injection of $2.0 \times 10^{12}$ genomic particles per ml (gp/ml) at 0.2 μl/min rate. CQ (40 mg/kg/day) or 4A7C-301 (5 mg/kg/day) administration was started 4 weeks after AAV vector injections and continued for 5 weeks before sacrifice at 8 weeks after surgery.

## Animal behaviors

**Rotarod.** Mice were pre-trained on an automated 5-lane rotarod unit (10 rpm, 3 min) for 2 days and the baseline was measured (10 rpm, 90 sec) before the rotarod test (3 days before the MPTP injection or the first rotarod test at week 2). Rotarod measurements were performed using an accelerating protocol, accelerated smoothly from 2 to 30 rpm for 5 min. Time on the rotating rod was measured automatically by placing a trip switch under the floor beneath the rotating drum.

**Pole test.** Mice were trained to descend a vertical metal rod wrapped with rough tape (80 cm high, 12 mm diameter) once per day for 2 days, and then the baseline was measured before the pole test (3 days before the MPTP injection or the first pole test at week 2). Mice were placed head-upward on the pole and total descent time was recorded.

**Cylinder test.** Mice were placed in a transparent plastic cylinder (15.5 cm high, 12.7 cm diameter) with a mirror behind it. Spontaneous activity was recorded and measured as counting the number of rearing events for 3 min session. A rear was considered only when the mouse touched the wall of the cylinder with its forelimb paws while standing on its hindlimbs (vertical movement) to count meaningful physiological movements.

**Olfactory discrimination.** The olfactory discrimination test was designed based on Prediger et al. [62]. Briefly, mice were placed in a chamber (40 (w) × 20 (d) × 20 (h) cm) segmented with two identical compartments containing fresh sawdust (non-familiar odor) on one side and unchanged sawdust (familiar odor; maintained for 3 days before the test) on the other side. Mice could choose between one or another area by freely passing through an open door, for 5 min. Mice movements were monitored with EthoVision XT version 7.0 (Noldus) and time (s) and distance (cm) traveled in each compartment were recorded and analyzed.

**AIMs scoring.** Abnormal involuntary movements (AIMs) were scored using the developed mouse dyskinesia rating scale as described in Sebastianutto et al. [48]. Briefly, 4 mice were recorded simultaneously with a Swann HD Security System (B&H Photo-Video Inc.) for 2 min per session, every 20 min during the 140 min following drug injections (i.e., total 7 sessions), on every other or third day (Day 2, 4, 7, 9, 11, 13, and 16). Dyskinetic movements were scored for 3 subtypes of basic AIMs including axial, forelimb and orolingual AIMs, based on the severity scale from 0 to 4 which measures the duration of AIMs (0 = no dyskinesia; 1 = display dyskinesia for <50% of the observation time; 2 = display dyskinesia for >50% of the observation time; 3 = display dyskinesia for the entire observation period but ceased upon external stimuli; 4 = display continuous and unceasing dyskinesia). To get a better and more sensitive validation of mice dyskinesia, we included an amplitude AIMs score, which rates the degree of deviation of body part or muscle position from its resting position on a 0-4 scale as previously described[8]. Finally, global AIMs score was calculated by multiplying basic AIMs and amplitude AIMs scores for each subtype on each session. Data points were plotted as the grand total of individual AIMs score on each testing day.

## Immunohistochemistry (IHC) and immunofluorescence (IF)

Perfused mouse brain tissues were cut into 30 μm thickness using a Leica cryostat (CM 1950). After incubation in blocking solution (1% horse serum (HS) and 0.3% Triton X-100 in PBS (pH 7.4)), brain sections were incubated with the following primary antibodies: TH (1:1,000; Millipore, MAB318), Iba-1 (1:2000; Abcam, ab178846), Nurr1 (1:1,000), phospho-S129 (1:5,000; ab51253), GFP (1:500; Aves Labs, GFP-1010). For IHC, secondary biotinylated anti-mouse (1:200; Vector Laboratories, BA-2001) or anti-rabbit (1:1,000; BA-1100) IgG antibodies were used followed by series of avidin-biotin complex (ABC; Vector Laboratories) and 3,3'-diaminobenzidine (DAB; Sigma-Aldrich) reactions according to the manufacturer's instruction. For IF, Alexa Fluor® 568 anti-mouse (1:500; Life Technologies, A11031) and Alexa Fluor® 488 anti-chicken (1:500; Life Technologies, A11039) secondary antibodies were used. Images were acquired using a KEYENCE microscope and quantified as the number or optical density of immunoreactive cells using ImageJ software.

## Stereological analysis

The numbers of $TH^+$ DA neurons, $NeuN^+$ neurons, pS129 $\alpha Syn^+$ cells in the SN and Iba-1$^+$ microglia in the SN and striatum were stereologically counted in six coronal sections (30 μm thickness) taken every 6th section (180 μm intervals) for each brain. The boundaries of the SN and striatum were delimited at low magnification (4X) to estimate each total area. Immunoreactive cells were counted with images taken at high magnification (×40) by a single blinded investigator. The group mean of the total number of immunoreactive cells from each brain was transformed as percentage of the number in the control group (%). For AAV-αSyn-induced mice, the group mean of the total number of cells in the ipsilateral side was transformed as a percentage of that observed in the contralateral side.

## Enzyme-linked immunosorbent assay (ELISA)

The endogenous dopamine levels in the brain tissue extracts from CQ or 4A7C-301 treated MPTP- or AAV-αSyn-injected mice were determined using a mouse dopamine ELISA Kit (MyBiosource), as previously described[33]. For MPTP-induced mice, the left hemisphere of each group (n = 5) was used for analysis and compared to the vehicle-treated group (VEH). For AAV-αSyn-injected mouse model, the viral vector injected side (left hemisphere; ipsilateral) of each group (n = 3) was compared to the non-injected side (right hemisphere; contralateral).

## BBB permeability assay

Final dose of 2 mg/mL 4A7C-301 dissolved in saline was orally administrated to SD rats (male, 7-9 weeks, n = 3 per group). To measure the 4A7C-301 concentration in the brain, rats were fully exsanguinated to remove residual blood prior to brain collection. Brain and blood samples were collected at 0.25, 0.5, 1, 2, 4, 6, 8, and 24 h after administration. A robust LC-MS/MS method was developed with the lower limit of quantification (LLOQ) of 1 ng/mL and 0.5 ng/mL for the quantification of 4A7C-301 in plasma and brain, respectively. The accuracy of all accepted QC samples of 4A7C-301 at low, middle, and high concentration levels met the acceptance criteria. Isolated brains were homogenized in PBS as brain weight (g) to PBS volume (mL) ratio 1:3. The desired serial concentrations of working solutions were achieved by diluting stock solution of analyte with 50% acetonitrile in water solution. 5 μL of working solutions (5, 10, 20, 50, 100, 500, 1000, 5000 ng/mL) were added to 50 μl of male SD rat blank brain homogenate to achieve calibration standards of 0.5 ~ 500 ng/mL (0.5, 1, 2, 5, 10, 50, 100, 500 ng/mL) in a total volume of 55 μl. Four quality control samples at 1.5 ng/mL, 3 ng/mL, 50 ng/mL and 400 ng/mL for brain homogenate were prepared independently of those used for the calibration curves. These QC samples were prepared on the day of analysis in the same way as calibration standards. 55 μl of standards, 55 μl of QC samples and 55 μl of unknown samples (50 μL of brain homogenate with 5 μL of blank solution) were added to 200 μl of acetonitrile containing IS mixture for precipitating protein, respectively. Then the samples were vortexed for 30 seconds. After centrifugation at 4 °C, 4000 x g for 15 min, the supernatant was diluted 5 times with water. 25 μl of diluted supernatant was injected into the LC-MS/MS system (AB API 5500 + LC-MS/MS instruments (Serial No. EX224682008)) for quantitative analysis. To measure the 4A7C-301 concentration in the plasma, serum samples were obtained from the collected blood and analyzed by LC-MS/MS system as indicated above.

## Statistics

GraphPad Prism (Version 8.0.2) was used for all statistical analyses and the specific tests used are described in the figure legends. Data were compared between two groups or within a group using Student's *t* test and compared among several groups using one- or two-way ANOVA followed by *post hoc* tests with Tukey's, Dunnett's or Bonferroni's multiple comparisons. Data are represented as mean ± s.e.m., and statistical significance was accepted for $P < 0.05$.

## Reporting summary

Further information on research design is available in the Nature Portfolio Reporting Summary linked to this article.

## Data availability

Data are available upon request to the corresponding author. Source data are provided with this paper.

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

## Acknowledgements

This work was supported by NIH grants (NS127391 and OD024622; to K.-S.K.), the Parkinson's Cell Therapy Research Fund at McLean Hospital (to K.-S.K.) and by SERB (Science and Engineering Research Board) grant (EMR/2014/001127; to D.S.R.) from the government of India. D.S.R. also thanks USIC-CIF, University of Delhi for analytical data of chemical compounds.

## Author contributions

W.K., M.T., D.S.R. and K.-S.K. designed the study. W. K. performed most of the experiments and analyses. M.T., S.V., S.K.K., R.K., A.T., S.M. and G.S. performed and confirmed synthetic procedures. C.K. established high throughput assay systems. Y.C. performed mitochondrial activity assay. S.K. generated lentiviruses. M.F., E.S., Y.-B.K., Y.K. and K.B. contributed to in vivo experiments. Y.-S.K. provided AAV vectors. Y.-S.K. and B.C. provided intellectual input. W.K., M.T., D.S.R., and K.-S.K. wrote the manuscript, with discussion and feedback from all co-authors.

## Competing interests

W.K., M.T., S.V., S.K.K., R.K., A.T., D.S.R., and K.-S.K. are co-inventors of a pending U.S. utility patent application no. 18/013,155 that is assigned to the McLean Hospital Corporation and the University of Delhi. The patent application no. 18/013,155 describes and claims compounds including SPV-94 (4A7C-301) that are described in this manuscript and methods of using these compounds to treat neurodegenerative diseases. K.-S.K. is a co-founder of NurrOn Pharmaceutical, Inc., which has rights to develop compounds disclosed in this manuscript under a licensing agreement with The McLean Hospital Corporation. The remaining authors have no conflicts of interest.
