## [Peer Review File · Nature Communications]

An optimized Nurr1 agonist provides disease-modifying effects
in Parkinson's disease modelsEditorial Note: This manuscript has been previously reviewed at another journal that is not operating a transparent peer review scheme. This document only contains reviewer comments and rebuttal letters for versions considered at Nature Communications.

REVIEWER COMMENTS

Reviewer #1 (Remarks to the Author):

Kim and colleagues have significantly improved the manuscript and addressed several issues. Details on compound development have been added, selectivity has at least been studied for related receptors and evidence for direct binding to Nurr1 has been improved. The study will be an important addition to characterizing Nurr1 as potential target in PD. However, a few further points need attention before publication.

- The title of this manuscript still is a stretch. The Nurr1 ligand presented here is an improvement over what is available but terming this ligand "optimal" is inappropriate due to several unclear aspects of its activity (as elaborated also in the authors' response letter). The term "optimal agonist" should be changed to "optimized agonist" or something similar that appropriately characterizes the compound.
- The reported compound may become a tool to study the biology of Nurr1 used by others in the field. Therefore, it is critical that the authors clearly present the limitations of this compound (as elaborated in the response letter) in the discussion. This should at least include i) the unclear effects on Nurr1 protein levels, ii) the broad variance of compound potency in different in vitro assays and the lack of a Kd value, iii) the unclear mode of action according to the authors' speculation of at least two mechanisms, iv) the still very limited data on compound selectivity, and v) the unclear binding site.
- Selectivity assays (at least nur77) lack references/positive controls.
- Further conclusions should be drawn from the mutagenesis study to derive a binding site hypothesis.

Reviewer #2 (Remarks to the Author):

Summary: Kim et al. have provided additional support and addressed comments raised in a previous round of review of the manuscript. The new compound they characterize, 4A7C-301, shows interesting properties. Even though the compound was derived through chemical modification from the 4A7C core present in chloroquine, it shows somewhat different chemistry. While chloroquine inhibits autophagy, 4A7C-301 appears to protect autophagy. This is an important distinction since autophagy inhibition by chloroquine has been shown to dramatically accelerate the deposition of α -synuclein pathology (PMID: 28611062). It is therefore surprising that in most of the assays, chloroquine and 4A7C-301 have similar effects. This point warrants further discussion in the manuscript.

The authors have thoroughly addressed previous comments. The major concern remaining is that these compounds likely have several effects that are not dependent on Nurr1 targeting. It will be important to address off-target effects and potential toxicities of these compounds in future work. For the current manuscript, it is advisable that the discussion of the use of these compounds as therapeutic agents is appropriately tempered.

Major Points:

1. In the previous round of review, the authors were asked to make clear what their experimental repeats referred to. They noted that all repeats were independent biological repeats. However, this is not correct. For example, Fig 3b, 3d; Extended Fig. 7b, 7d: 30 cells or 15 cells used for the repeats here cannot be considered biologically independent samples.

Minor Points:

1. Studies in which fibril injections were used (PMID: 23161999, 28668282) were inappropriately conflated with α -synuclein overexpression in the author rebuttal. These are very different systems, and it is still highly surprising that the authors see olfactory deficits in their animals.
2. The authors incorrectly assert that they are unable to change the color of images due to the fluorophores available to them. Fluorescence images are typically collected in monochrome and are pseudocolored by the instrument or by the user, and therefore the reported color is not dependent on the specific fluorophore used. If the authors prefer the current colors, and the journal agrees, that is up to them, but it is incorrect to convey that changing the colors is not possible.

Reviewer #3 (Remarks to the Author):

Review of transferred Nature Communications manuscript NCOMMS-22-53914-T

This is a revised manuscript that was transferred to Nature Communications for consideration. As highlighted in the first review, the authors report the results of an extensive medicinal chemistry search in which over 570 4A7C-derivatives were generated and characterized for their ability to activate Nurr1. This review focuses on the issues that were identified in the first review.

- 1) Despite the author's rebuttal the only real conceptual advance is the identification of a new Nurr1 activator. Nurr1 is well known to play a role in PD models and it is somewhat disingenuous to imply that it is a conceptual advance.
- 2) It is still not clear whether the authors performed stereology to assess DA neuronal loss. Searching the document for stereology was negative. Scientific rigor dictates that stereology must be used in these types of studies. Although the authors performed immunostaining for NeuN, the lack of stereology does not instill confidence in the findings.
- 3) A simple request was made to determine whether 4A7C-301 affected the metabolism of MPTP. Instead the authors showed 4A7C-301 the basal respiration and ATP turnover rate were decreased by MPP+ in Nurr1 KD condition, even in the presence of pre-treated 4A7C-301. These data do not shed any light on whether 4A7C-301 affects the metabolism of MPTP to MPP+. Since MPP+ is the critical determinant of MPTP toxicity, it still remains unknown whether 4A7C-301 protective effect is simply due to reducing the availability of MPP+.

In sum, the current version of the revised paper falls short in addressing the major concerns that were identified in the first version.

REVIEWER COMMENTS

Reviewer #1 (Remarks to the Author):

Kim and colleagues have significantly improved the manuscript and addressed several issues. Details on compound development have been added, selectivity has at least been studied for related receptors and evidence for direct binding to Nurr1 has been improved. The study will be an important addition to characterizing Nurr1 as potential target in PD. However, a few further points need attention before publication.

[Response] We greatly appreciate the reviewer's positive evaluation stating that "Kim and colleagues have significantly improved the manuscript and addressed several issues" and "the study will be an important addition to characterizing Nurr1 as potential target in PD". At the same time, the reviewer pointed out a few important issues that need attention before publication, which are comprehensively addressed, as described below.

- The title of this manuscript still is a stretch. The Nurr1 ligand presented here is an improvement over what is available but terming this ligand "optimal" is inappropriate due to several unclear aspects of its activity (as elaborated also in the authors' response letter). The term "optimal agonist" should be changed to "optimized agonist" or something similar that appropriately characterizes the compound.

[Response] We thank the reviewer for this suggestion. Based on this suggestion, we changed the title to "An optimized Nurr1 agonist provides disease-modifying effects in Parkinson's disease models". In the same manner, we changed the text to "optimized agonist(s)" in the Discussion (line 410, page 12; lines 463-464, page 13) and in the Figure legend (line 467, page 15).

- The reported compound may become a tool to study the biology of Nurr1 used by others in the field. Therefore, it is critical that the authors clearly present the limitations of this compound (as elaborated in the response letter) in the discussion. This should at least include i) the unclear effects on Nurr1 protein levels, ii) the broad variance of compound potency in different in vitro assays and the lack of a K_d value, iii) the unclear mode of action according to the authors' speculation of at least two mechanisms, iv) the still very limited data on compound selectivity, and v) the unclear binding site.

[Response] We agree with the reviewer. The reported compound may become a useful tool to study the biology of Nurr1 but may still have limitations. Accordingly, we included those potential limitations of the compound along with the comments on necessary future studies to determine clinically applicable drug candidate in the Discussion section of the revised manuscript (lines 457-465, page 13).

In addition, we changed the last sentence of the Abstract "These substantial disease-modifying properties of 4A7C-301 may warrant its clinical evaluation for the treatment of patients with PD" to "These substantial disease-modifying properties of 4A7C-301 may warrant clinical evaluation of this or analogous compounds for the treatment of patients with PD".

- Selectivity assays (at least nur77) lack references/positive controls.

[Response] To address the reviewer's comment, we performed additional reporter assays, including using cytosporone B as a positive Nur77 agonist (Zhan *et al.*, Nat Chem Biol 2008 [PMID: 18690216]) together with vehicle control. In addition, we tested another positive control prostaglandin A1 (PGA1) for both Nurr1 and Nor1, as recently described in the Rajan *et al.* study (Nat Chem Biol 2020 [PMID: 32451509]). As shown below, cytosporone B, but not CQ or 4A7C-301, robustly and selectively activated the transcriptional activity of Nur77. In the case of Nor1, its transcriptional activity was enhanced by both 4A7C-301 and PGA1. These new data and related references have been added in the revised manuscript (lines 180-183, pages 6-7; Fig. 1f). Consistent with our previous results, our new data further support the notion that 4A7C-301 selectively activates the transcriptional activity of Nurr1, but not that of Nur77, over Nor1.

Response Fig. 1. Effect of CQ (100 μ M) and 4A7C-301 (20 μ M) on the transcriptional activities of NR4A subfamily members in SK-N-BE(2)C cells. PGA1 (10 μ M) and cytosporone B (10 μ M) were used as positive controls for Nor1 and Nur77 activation, respectively. Data are mean \pm s.e.m. $n = 3$ independent samples (replaced in Fig. 1f).

- Further conclusions should be drawn from the mutagenesis study to derive a binding site hypothesis.

[Response] Based on the reviewer's suggestion, we drew further conclusions from the mutagenesis studies as follows:

We selected specific residues in Nurr1-LBD for the mutagenesis study based on nuclear magnetic resonance (NMR) titration data from our previous published studies (Kim *et al.*, Proc Nat Acad Sci U S A, 2015 [PMID: 26124091]; Park *et al.*, Sci Rep 2019 [PMID: 31664129]; Rajan *et al.*, Nat Chem Biol 2020 [PMID: 32451509]). Notably, although these studies demonstrated that AQ (Kim *et al.*, Proc Nat Acad Sci U S A, 2015 [PMID: 26124091]), CQ (Park *et al.*, Sci Rep 2019 [PMID: 31664129]), and PGA1/PGE1 (Rajan *et al.*, Nat Chem Biol 2020 [PMID: 32451509]) interact with the ligand binding domain of Nurr1, specific interacting residues were significantly different between these agonists. Because we selected 4A7C-301 among CQ derivatives, we chose residues S441, I573, A586, I588, K590, L593, D594, T595, L596 and F598 which were previously identified as interaction sites with CQ (Park *et al.*, Sci Rep 2019 [PMID: 31664129]). Significant reduction of transcriptional activity was observed in reporter constructs with mutations at most (I573, I588, L593, D594, T595, L596 and F598) but not all these residues in similar patterns with CQ and 4A7C-301 (Fig. 1g). Those findings confirm these sites as critical for interaction and activation by CQ/4A7C-301, and further reveal that both CQ and 4A7C-301 activate Nurr1 via direct binding to Nurr1-LBD. Notably, mutations at certain residues (e.g., S441, K590, L593 and D594) did not affect the basal transcriptional activity but decreased CQ- or 4A7C-301-induced Nurr1 transcriptional activity compared to the wild-type reporter construct. When the effects of mutation at these residues were further tested by treating the transfected cells with 4A7C-301 as well as with PGA1 and AQ, Nurr1 transcriptional activation by 4A7C-301, but not that by AQ or PGA1, was significantly diminished in these mutant constructs. These results support the conclusion that these residues are 4A7C-301 (and CQ)-specific (Extended Data Fig. 2). We added this discussion and modified the site-directed mutagenesis part in the revised manuscript (lines 189-206, page 7).

Reviewer #2 (Remarks to the Author):

Summary: Kim et al. have provided additional support and addressed comments raised in a previous round of review of the manuscript. The new compound they characterize, 4A7C-301, shows interesting properties.

[Response] We greatly appreciate the reviewer's positive evaluation.

Even though the compound was derived through chemical modification from the 4A7C core present in chloroquine, it shows somewhat different chemistry. While chloroquine inhibits autophagy, 4A7C-301 appears to protect autophagy. This is an important distinction since autophagy inhibition by chloroquine has been shown to dramatically accelerate the deposition of α -synuclein pathology (PMID: 28611062). It is therefore surprising that in most of the assays, chloroquine and 4A7C-301 have similar effects. This point warranted further discussion in the manuscript.

[Response] We agree with the reviewer that CQ's autophagy inhibition and 4A7C-301's autophagy protection is an important distinction because autophagy inhibition by CQ dramatically accelerates the deposition of α -synuclein pathology (Karpowicz *et al.*, J Biol Chem 2017 [PMID: 28611062]). This is exactly why we have extensively investigated the differential effects of CQ and 4A7C-301 on autophagy progression (Fig. 3a-g; Extended Fig. 7a-j). As the reviewer stated, CQ and 4A7C-301 show somewhat different chemistry, although they share a core structure, 4-amino-7-chloroquinoline, and exhibited similar effects in most of the assays. Thus, we speculate that, while the same core structure allows both CQ and 4A7C-301 to bind Nurr1 (related to the putative SAR), their different side chain structures underlie their distinct effects on autophagy. In support of this interpretation, we found that both CQ and bafilomycin A₁ (BafA₁), well-known autophagy inhibitors, significantly increased lysosomal pH in both dopaminergic N27-A (Fig. 3c,d) and HeLa cells (Extended Data Fig. 7c,d). In sharp contrast, 4A7C-301 did not affect lysosomal pH at all, suggesting possible different mechanisms underlying CQ/4A7C-301's distinct effects on autophagy.

Based on the reviewer's suggestion, we added this discussion, along with the references noted above, in the revised manuscript (lines 433-440, page 13).

The authors have thoroughly addressed previous comments. The major concern remaining is that these compounds likely have several effects that are not dependent on Nurr1 targeting. It will be important to address off-target effects and potential toxicities of these compounds in future work. For the current manuscript, it is advisable that the discussion of the use of these compounds as therapeutic agents is appropriately tempered.

[Response] We greatly appreciate the reviewer's insightful suggestion, which is also shared by other reviewers. Accordingly, we included the potential limitations of the compound and the necessary future studies to determine the clinically appropriate drug candidate; these comments appear in the Discussion section of the revised manuscript (lines 457-465, page 13). In addition, we changed the last sentence of the Abstract "These substantial disease-modifying properties of 4A7C-301 may warrant its clinical evaluation for the treatment of patients with PD" to "These substantial disease-modifying properties of 4A7C-301 may warrant clinical evaluation of this or analogous compounds for the treatment of patients with PD".

Major Points:

1. In the previous round of review, the authors were asked to make clear what their experimental repeats referred to. They noted that all repeats were independent biological repeats. However, this is not correct. For example, Fig 3b, 3d; Extended Fig. 7b, 7d: 30 cells or 15 cells used for the repeats here cannot be considered biologically independent samples.

[Response] We apologize that our explanation was unclear. For autophagy analyses, we transfected HeLa and N27-A cells with tandem mRFP-GFP-LC3 plasmid using Lipofectamine 2000 in 24-well plates. After 24 hrs, cells were incubated in starvation medium (Earle's Balanced Salt Solution, EBSS; Lonza) containing bafilomycin A₁ (BafA₁, 10 nM), CQ (20 μ M), or 4A7C-301 (1 μ M) for 6 hrs and then fixed with 4% formaldehyde for fluorescence detection. Images were taken using a KEYENCE microscope and red (mRFP) LC3 dots and yellow (mRFP+GFP) LC3 dots were counted from three independent wells for each treatment group. 10 cells were randomly picked

and counted from each well and a total of 30 cells were analyzed for each treatment in both HeLa and N27-A cells. Thus, three wells for each treatment group are indeed biological replicates. We revised the Methods section (lines 806-812, page 26; lines 819-820, page 27) and the related legends of Fig. 3 (lines 543-548, page 17) and Extended Data Fig. 7 (lines 198-203, page 11).

In addition, to further ensure the reproducibility of our data, we performed additional independent experiments for both HeLa and N27-A cells to fulfill the statistics from “biological independent samples” in different experiments. As shown below (Response Fig. 2 and 3), these new data showed identical patterns to the original data. Furthermore, since we obtained similar patterns of data from two different cell lines, it strongly supports our observations and conclusion about the different effects of CQ and 4A7C-301 on autophagy regulation.

These new data from an additional independent experiment, shown below, are not included in our revised manuscript. However, if the reviewer requests to include them in the revised manuscript, we will be happy to do so.

Response Fig. 2. Autophagolysosome (APL) formation assay and lysosomal pH detection in N27-A cells. (a) Number of yellow LC3 dots and red LC3 dots per cell counted from 10 random cells in each well from triplicates for each condition (total of 30 cells per each group). Two-way ANOVA, Tukey's multiple comparisons. Data are mean \pm s.e.m. (b) Quantification from 5 random cells in each well from triplicates for each condition (total of 15 cells per each group). One-way ANOVA, Tukey's multiple comparisons. Data are mean \pm s.e.m.

Response Fig. 3. APL formation assay and lysosomal pH detection in HeLa cells. (a) Number of yellow LC3 dots and red LC3 dots per cell counted from 10 random cells in each well from triplicates for each condition (total of 30 cells per each group). Two-way ANOVA, Tukey's multiple comparisons. Data are mean \pm s.e.m. (b) Quantification from 5 random cells in each well from triplicates for each condition (total of 15 cells per each group). One-way ANOVA, Tukey's multiple comparisons. Data are mean \pm s.e.m.

Minor Points:

1. Studies in which fibril injections were used (PMID: 23161999, 28668282) were inappropriately conflated with α -synuclein overexpression in the author rebuttal. These are very different systems, and it is still highly surprising that the authors see olfactory deficits in their animals.

[Response] We apologize for our oversight in the previous rebuttal letter that fibril injection studies were inappropriately conflated with α -synuclein overexpression and agree that they are very different systems.

As the reviewer states, it is surprising and interesting that we observed olfactory deficits in both MPTP-induced and α -synuclein overexpression models and that 4A7C-301 and CQ significantly rescued those deficits. Notably, recent studies similarly observed olfactory dysfunction using animal models with both α -synuclein overexpression (Martin-Lopez *et al.*, 2023 [PMID: 36596700]) and PFF injection (Uemura *et al.*, 2021 [PMID: 33547846]). However, the molecular mechanisms underlying olfactory dysfunction in these animal models and its amelioration by Nurr1 agonists are still not clearly understood and await further investigation. We added discussion of these observations in the revised manuscript along with relevant references (lines 446-452, page 13).

2. The authors incorrectly assert that they are unable to change the color of images due to the fluorophores available to them. Fluorescence images are typically collected in monochrome and are pseudocolored by the instrument or by the user, and therefore the reported color is not dependent on the specific fluorophore used. If the authors prefer the current colors, and the journal agrees, that is up to them, but it is incorrect to convey that changing the colors is not possible.

[Response] Based on the reviewer's comment, we changed immunofluorescence images in Fig. 2c, Fig. 4i and Fig. 5i to a green-blue scheme from a green-red scheme (see Response Fig. 4-6). For the autophagolysosome (APL) formation assay, we kept the dot colors as red-green (and yellow as its merged image) scheme since the red-green mode is generally used based on its assay construct, mRFP-GFP-LC3, as shown in previous studies (e.g., Ni *et al.*, *Autophagy* 2011 [PMID: 21107021]; Chen *et al.*, *Mol Cell* 2011 [PMID: 22342342]; Li *et al.*, *Cell Death Dis* 2020 [PMID: 32980859]). Similarly, we kept the red-yellow-green color range for the indication of acidic to basic status of lysosomal pH using LysoSensor™ Yellow/Blue DND-160, as described in the previous literatures (e.g., Ni *et al.*, *Autophagy* 2011 [PMID: 21107021]; Ma *et al.*, *Front Cell Develop Biol* 2017 [PMID: 28871281]).

Response Fig. 4. Representative immunofluorescence images of TH and Iba-1 staining in mVM-glia co-culture in a blue-green color scheme (replaced in Fig. 2c).

Response Fig. 5. Representative immunofluorescence images of TH and NeuN staining in the SNpc of MPTP-induced mice in a blue-green color scheme (replaced in Fig. 4i).

Response Fig. 6. Representative immunofluorescence images of TH and NeuN staining in the SNpc of AAV- α Syn-induced mice in a blue-green color scheme (replaced in Fig. 5i).

Reviewer #3 (Remarks to the Author):

Review of transferred Nature Communications manuscript NCOMMS-22-53914-T

This is a revised manuscript that was transferred to Nature Communications for consideration. As highlighted in the first review, the authors report the results of an extensive medicinal chemistry search in which over 570 4A7C-derivatives were generated and characterized for their ability to activate Nurr1. This review focuses on the issues that were identified in the first review.

1) Despite the author's rebuttal the only real conceptual advance is the identification of a new Nurr1 activator. Nurr1 is well known to play a role in PD models and it is somewhat disingenuous to imply that it is a conceptual advance.

[Response] We agree with the reviewer that the conceptual advance is the identification of a new Nurr1 activator, and our work serves as foundational preclinical studies exploring the efficacy of 4A7C-301 as a disease modifying agent for the treatment of PD. As the reviewer states, Nurr1 is well known to be an important player in PD and animal models of PD. Our study is by nature translational research for therapeutic development. That said, we believe that our study has substantial value in that translational sphere.

2) It is still not clear whether the authors performed stereology to assess DA neuronal loss. Searching the document for stereology was negative. Scientific rigor dictates that stereology must be used in these types of studies. Although the authors performed immunostaining for NeuN, the lack of stereology does not instill confidence in the findings.

[Response] We apologize for the lack of description of the stereological methods we used in our study. Indeed, we performed stereology to assess DA neuronal loss and protection associated with treatments with agonists. In the revised manuscript, we clarified our stereological quantification (lines 339-343, page 10; 373-380, page 11). In addition, we added detailed stereological analysis procedures in the Methods section (lines 913-922, page 29), as shown below.

Stereological analysis

The number of TH⁺ DA neurons, NeuN⁺ neurons, and pS129 α Syn⁺ cells in the SN and Iba-1⁺ microglia in the SN and striatum was stereologically counted in six coronal sections (30 μ m thickness) taken every 6th section (180 μ m intervals) for each brain. The boundaries of the SN and striatum were delimited at low magnification (4X) to estimate the area. Immunoreactive cells were counted with images taken at high magnification (40X) by a single blinded investigator. The group mean of the total number of immunoreactive cells from each brain was transformed as percentage of control group (%). For AAV- α Syn-induced mice, the group mean of the total number of cells from the ipsilateral side was transformed as a percentage of that observed in the contralateral side.

3) A simple request was made to determine whether 4A7C-301 affected the metabolism of MPTP. Instead the authors showed 4A7C-301 the basal respiration and ATP turnover rate were decreased by MPP⁺ in Nurr1 KD condition, even in the presence of pre-treated 4A7C-301. These data do not shed any light on whether 4A7C-301 affects the metabolism of MPTP to MPP⁺. Since MPP⁺ is the critical determinant of MPTP toxicity, it still remains unknown whether 4A7C-301 protective effect is simply due to reducing the availability of MPP⁺.

[Response] We apologize for our oversight to clarify this issue. In our sub-chronic MPTP regimen, we first injected each mouse with MPTP and at 6 hrs post-MPTP injection administered each compound (L-DOPA, CQ, or 4A7C-301). These procedures were based on those used in previous studies which investigated MPTP/MPP⁺ kinetics in various brain regions of C57BL/6 mice (e.g., Fornai *et al.*, J Pharmacol Exp Ther 1997 [PMID: 9336313]). In these studies, MPTP was completely metabolized to MPP⁺ within 1 hr in both the striatum and the substantia nigra (SN), as shown below, in Reference Table 1. The active metabolite (MPP⁺) level peaks at around 30 min after MPTP injection and decreases time-dependently. Based on this finding, previous MPTP-based animal studies have chosen a time window of 4-14 hrs post-MPTP injection for testing drug effects in PD animal models (Jackson-Lewis and Przedborski, Nat Prot 2007 [PMID: 17401348]; Brynskikh *et al.*, Nanomedicine 2010 [PMID: 20394532]; Feng *et al.*, Neuropharmacology 2018 [PMID: 29462693]; Gottschalk *et al.*, Neurobiol Dis 2021 [PMID: 33514677]).

Since we administered these drugs at 6 hrs post-MPTP injection, when MPTP is completely converted to MPP⁺, we believe that our findings did not result from our drug candidates causing reduced availability of MPP⁺.

In further support of this conclusion, we observed the same superior 4A7C-301 protective effect in *in vitro* assays where MPP⁺ was directly administered to cells (Fig. 2a,b; Extended Data Fig. 3e,f).

We thank the reviewer for asking for further clarification of this issue and clarified the experimental procedure in more detail in the Methods section (lines 837-839, page 27).

Reference Table 1. MPTP/MPP⁺ kinetics in striatum and SN of C57L/6 mice (modified from Fornai *et al.*, 1997)

		Striatum	SN
10 min	MPTP	109.1 ± 9.1	45.5 ± 7.5
	MPP ⁺	0.2 ± 0.2	21.8 ± 3.6
30 min	MPTP	/	20.9 ± 6.5
	MPP ⁺	46.3 ± 4.3	80.9 ± 12.8
1 h	MPTP	/	/
	MPP ⁺	80.1 ± 7.4	61.0 ± 5.8
2 h	MPP ⁺	75.4 ± 4.4	44.9 ± 6.1
4 h	MPP ⁺	44.1 ± 4.9	20.8 ± 2.7
6 h	MPP ⁺	14.5 ± 1.3	12.1 ± 1.4
12 h	MPP ⁺	6.9 ± 0.5	/

In sum, the current version of the revised paper falls short in addressing the major concerns that were identified in the first version.

[Response] We hope that our responses, detailed above, now successfully address the reviewer's concerns. In addition, this reviewer previously raised another issue regarding the potential limitation of 4A7C-301 as a disease modifying agent in PD. Together with this concern, we have included a discussion of the potential limitations of the current compound along with the necessary future studies needed to determine a clinically applicable drug candidate. These points are in the Discussion section of the revised manuscript (lines 457-465, page 13).

REVIEWERS' COMMENTS

Reviewer #1 (Remarks to the Author):

The authors have convincingly addressed all comments. The manuscript appears suitable for publication. A few further points (as follows) should be considered but this does not require further peer-review.

- The acronym RA for retinoic acid should be defined in the caption of figures 1h and 1i. Additionally, the authors should clarify the motivation to use RA as negative control here.
- SD or S.E.M. should be reported for the IC50 values from binding assays (Figure 1h and 1i). Additionally, the dose-response of 4A7C-301 is incomplete. I would recommend improving/completing this data set.
- The resolution of some subfigures of Fig. 5 is poor.

Reviewer #2 (Remarks to the Author):

Kim et al. have provided a thorough response to previous reviewer comments. Most of my major concerns have been addressed and the limitations of the study, as noted by several reviewers, have been acknowledged by the authors and incorporated into the manuscript. In assessing the responses, one final point arose which should be addressed prior to publication.

1. The number of experimental repeats has now been more thoroughly explained. However, this has made clear that not all experiments have had appropriate statistics applied. For example, in experiments where 10 cells were picked from 3 independent wells, the 10 cells cannot be considered independent replicates since their phenotypes are directly interrelated. Therefore, an ANOVA is not an appropriate statistical test since this test assumes each observation is independent. Authors should either only consider independent replicates (n=3, not n=30) or use a statistical test which can appropriately account for the nested data structure in these experiments.

Reviewer #3 (Remarks to the Author):

This is resubmission of a revised manuscript. It is significantly improved including changing the focus from a conceptual advance to potential translational importance. The authors also clarified and highlighted the use of stereology in their assessments. A big issue still remains as it is not clear if the authors actually measured MPP+ levels in MPTP treated mice +/- (4A7C-301). By not measuring and actually comparing MPP+ levels in the different treatments it is impossible to conclude that 4A7C-301 does not affect the metabolism of MPTP. Maybe I am missing these data, but I searched the entire manuscript with the term "MPP+" and I could not find where the authors had actually measured the conversion of MPTP to MPP+. There are a lot of MPP+ experiments, but these do not address the key question on the metabolism of MPTP to MPP+ and the levels of MPP+ achieved in the brain in these experiments. Thus it is still not known whether 4A7C-301 is decreasing the brain levels of MPP+. In other words if 4A7C-301 affects the brain levels of MPP+ it would fully account for its protective affect separate from its affects on Nurr1.

REVIEWERS' COMMENTS

Reviewer #1 (Remarks to the Author):

The authors have convincingly addressed all comments. The manuscript appears suitable for publication. A few further points (as follows) should be considered but this does not require further peer-review.

[Response] We greatly appreciate the reviewer for the very positive statement and suggesting a few further points. As shown below, we fully addressed these additional points.

- The acronym RA for retinoic acid should be defined in the caption of figures 1h and 1i. Additionally, the authors should clarify the motivation to use RA as negative control here.

[Response] We added the definition of RA and the purpose for its use in the corresponding legend (page 29, lines 1024-1025).

- SD or S.E.M. should be reported for the IC₅₀ values from binding assays (Figure 1h and 1i). Additionally, the dose-response of 4A7C-301 is incomplete. I would recommend improving/completing this data set.

[Response] Based on the reviewer's suggestion, we included the SD for the IC₅₀ values from binding assays (page 7, lines 203 and 206).

- The resolution of some subfigures of Fig. 5 is poor.

[Response] We replaced Fig. 5c, d with new figures with higher resolution.

Reviewer #2 (Remarks to the Author):

Kim et al. have provided a thorough response to previous reviewer comments. Most of my major concerns have been addressed and the limitations of the study, as noted by several reviewers, have been acknowledged by the authors and incorporated into the manuscript. In assessing the responses, one final point arose which should be addressed prior to publication.

[Response] We greatly appreciate the reviewer for the very positive statement and for suggesting one final point which we fully addressed, as shown below.

1. The number of experimental repeats has now been more thoroughly explained. However, this has made clear that not all experiments have had appropriate statistics applied. For example, in experiments where 10 cells were picked from 3 independent wells, the 10 cells cannot be considered independent replicates since their phenotypes are directly interrelated. Therefore, an ANOVA is not an appropriate statistical test since this test assumes each observation is independent. Authors should either only consider independent replicates ($n=3$, not $n=30$) or use a statistical test which can appropriately account for the nested data structure in these experiments.

[Response] Based on the reviewer's suggestion, we have changed the statistical tests for Fig. 3b, d and Extended Data Fig. 7b, d (which is Supplementary Fig. 7b, d in the revised version) to two-tailed unpaired t -test and revised the corresponding legends (page 31, lines 1122 and 1125; Supplementary Information file, page 12, lines 232 and 235).

Reviewer #3 (Remarks to the Author):

This is resubmission of a revised manuscript. It is significantly improved including changing the focus from a conceptual advance to potential translational importance. The authors also clarified and highlighted the use of stereology in their assessments.

[Response] We greatly appreciate the reviewer for the positive statement.

A big issue still remains as it is not clear if the authors actually measured MPP⁺ levels in MPTP treated mice +/- (4A7C-301). By not measuring and actually comparing MPP⁺ levels in the different treatments it is impossible to conclude that 4A7C-301 does not affect the metabolism of MPTP. Maybe I am missing these data, but I searched the entire manuscript with the term "MPP⁺" and I could not find where the authors had actually measured the conversion of MPTP to MPP⁺. There are a lot of MPP⁺ experiments, but these do not address the key question on the metabolism of MPTP to MPP⁺ and the levels of MPP⁺ achieved in the brain in these experiments. Thus it is still not known whether 4A7C-301 is decreasing the brain levels of MPP⁺. In other words if 4A7C-301 affects the brain levels of MPP⁺ it would fully account for its protective affect separate from its affects on Nurr1.

[Response] Based on the published data that we provided in our previous response (replicated below), MPTP is completely metabolized to MPP⁺ within 1 hr of administration in both the striatum and substantia nigra. Thus, we respectfully disagree with the reviewer's point that "it is still not known whether 4A7C-301 is decreasing the brain levels of MPP⁺". Since in our animal experiments we administered 4A7C-301 6 hours post-MPTP injection, there cannot be any MPTP remaining in either the striatum or in the substantia nigra. Thus, it is impossible for 4A7C-301 to change or interrupt the metabolism of MPTP to MPP⁺ (because there is no MPTP remaining at 6 hours post-MPTP).

Reference Table 1. MPTP/MPP⁺ kinetics in striatum and SN of C57L/6 mice (modified from Fornai *et al.*, 1997)

		Striatum	SN
10 min	MPTP	109.1 ± 9.1	45.5 ± 7.5
	MPP ⁺	0.2 ± 0.2	21.8 ± 3.6
30 min	MPTP	/	20.9 ± 6.5
	MPP ⁺	46.3 ± 4.3	80.9 ± 12.8
1 h	MPTP	/	/
	MPP ⁺	80.1 ± 7.4	61.0 ± 5.8
2 h	MPP ⁺	75.4 ± 4.4	44.9 ± 6.1
4 h	MPP ⁺	44.1 ± 4.9	20.8 ± 2.7
6 h	MPP ⁺	14.5 ± 1.3	12.1 ± 1.4
12 h	MPP ⁺	6.9 ± 0.5	/